# Exercise Improves Redox Homeostasis and Mitochondrial Function in White Adipose Tissue

**DOI:** 10.3390/antiox11091689

**Published:** 2022-08-29

**Authors:** Leonardo Matta, Caroline Coelho de Faria, Dahienne F. De Oliveira, Iris Soares Andrade, Niedson Correia Lima-Junior, Bianca Martins Gregório, Cristina Maeda Takiya, Andrea Claudia Freitas Ferreira, José Hamilton M. Nascimento, Denise Pires de Carvalho, Alexander Bartelt, Leonardo Maciel, Rodrigo Soares Fortunato

**Affiliations:** 1Health Science Center, Carlos Chagas Filho Institute of Biophysics, Federal University of Rio de Janeiro, 21941-590 Rio de Janeiro, Brazil; 2Institute for Cardiovascular Prevention, Klinikum der Universität München, Ludwig-Maximilians-University Munich, 80539 München, Germany; 3Institute for Diabetes and Cancer, Helmholtz Center Munich, Neuherberg, 85764 Munich, Germany; 4Urogenital Research Unit, Roberto Alcântara Gomes Institute of Biology, State University of Rio de Janeiro, 20511-010 Rio de Janeiro, Brazil; 5Multidisciplinary Center for Research in Biology (NUMPEX) Duque de Caxias Campus, Federal University of Rio de Janeiro, 25250-470 Rio de Janeiro, Brazil

**Keywords:** exercise, redox homeostasis, hormesis, adipose tissue

## Abstract

Exercise has beneficial effects on energy balance and also improves metabolic health independently of weight loss. Adipose tissue function is a critical denominator of a healthy metabolism but the adaptation of adipocytes in response to exercise is insufficiently well understood. We have previously shown that one aerobic exercise session was associated with increased expression of antioxidant and cytoprotective genes in white adipose tissue (WAT). In the present study, we evaluate the chronic effects of physical exercise on WAT redox homeostasis and mitochondrial function. Adult male Wistar rats were separated into two groups: a control group that did not exercise and a group that performed running exercise sessions on a treadmill for 30 min, 5 days per week for 9 weeks. Reactive oxygen species (ROS) generation, antioxidant enzyme activities, mitochondrial function, markers of oxidative stress and inflammation, and proteins related to DNA damage response were analyzed. In WAT from the exercise group, we found higher mitochondrial respiration in states I, II, and III of Complex I and Complex II, followed by an increase in ATP production, and the ROS/ATP ratio when compared to tissues from control rats. Regarding redox homeostasis, NADPH oxidase activity, protein carbonylation, and lipid peroxidation levels were lower in WAT from the exercise group when compared to control tissues. Moreover, antioxidant enzymatic activity, reduced glutathione/oxidized glutathione ratio, and total nuclear factor erythroid-2, like-2 (NFE2L2/NRF2) protein levels were higher in the exercise group compared to control. Finally, we found that exercise reduced the phosphorylation levels of H2AX histone (γH2AX), a central protein that contributes to genome stability through the signaling of DNA damage. In conclusion, our results show that chronic exercise modulates redox homeostasis in WAT, improving antioxidant capacity, and mitochondrial function. This hormetic remodeling of adipocyte redox balance points to improved adipocyte health and seems to be directly associated with the beneficial effects of exercise.

## 1. Introduction

Physical exercise is defined as a regular, guided, and programmed practice of training sessions to improve systemic fitness [1]. Each session of exercise disturbs body homeostasis leading to an adaptative response that is linked to the beneficial effects of exercise [2]. Several cellular and systemic effects induced by exercise seem to be related to hormesis [3]. The hormetic response is defined as a beneficial or stimulatory effect caused by exposure to low doses of an agent known to be toxic at higher doses (e.g., chemical or physical), generating adaptive beneficial effects on the physiology of an organism [2,3,4].

It has been demonstrated that the hormetic response induced by exercise involves redox signaling. Acutely, a bout of exercise stimulates the generation of reactive oxygen species (ROS) in several tissues, activating a multitude of signaling pathways related to stress adaptation in a redox-dependent manner [5], such as AMP-activated protein kinase (AMPK) [6], mitogen-activated protein kinase (MAP kinases) [7], nuclear factor kappa B (NFkB) [8], and nuclear factor erythroid 2-related factor 2 (NFE2L2, also known as NRF2) [9]. NRF2 has been identified as the master transcription factor for the regulation of genes related to enzymatic antioxidant defense and cytoprotection, modulating more than 250 genes [10]. Moderate and regular exercise activates the NRF2 pathway in tissues such as kidney [11], prostate [12], and testis [13], with a subsequent increase in the antioxidant defense [14,15]. Interestingly, exercise-induced NRF2 activation seems to improve mitochondrial function in muscle [16]. NRF2 increases mitochondrial biogenesis by the stimulation of peroxisome proliferator-activated receptor-gamma coactivator (PGC-1α) [17], and mitochondrial transcriptor factor A (TFam) gene expression [18]. NRF2 also improves mitochondrial membrane potential (∆ᴪm), -respiration [16], -fatty acid oxidation, -structural and -functional integrity of the matrix, and -ATP synthesis [19].

Chronic aerobic exercise has a wide range of effects on metabolism, including decreased lipid storage and fatty acid mobilization [20], improved mitochondrial function [21] and decreased expression of inflammatory adipokines [22] in white adipose tissue (WAT). However, little is known if this beneficial remodeling is linked to redox homeostasis. Redox homeostasis improvement induced by exercise is linked to a higher antioxidant defense capacity, which can be generally classified as non-enzymatic and enzymatic. The main enzymes responsible for the enzymatic antioxidant defense are SOD, CAT, and GPx. SOD is a metalloenzyme that catalyzes the dismutation of O_2_^●−^ into H_2_O_2_ that can be found as three isoforms: SOD1 found mainly in the cytosol, SOD2 in mitochondria, and SOD3 in the extracellular space [23]. GPx catalyzes the reduction of H_2_O_2_, dependent on the oxidation of GSH to GSSG, releasing H_2_O [24]. Finally, CAT is a cytoplasmic hemeprotein that catalyzes the reduction in H_2_O_2_ to water and O_2_ [25]. Taken together these enzymes neutralize pro-oxidative molecules, such as ROS. There are several pieces of evidence supporting the antioxidant effect of exercise in different tissues [26,27]. Townsend, L. K. et al. (2020) demonstrated that catalase overexpression reduced basal and exercise-induced ROS production in WAT [28]. Sakurai et al. (2009) showed that 9 weeks of aerobic exercise increased superoxide dismutase 2 expression, decreased NOX2 activity, and lipid peroxidation in retroperitoneal WAT [22]. However, a clear understanding of redox homeostasis and mitochondrial function in WAT is still missing.

Previously, we have found that rats undergoing one session of exercise presented a transient pro-oxidative state in retroperitoneal WAT, which was followed by an increase in NRF2-mediated antioxidants and cytoprotective gene expression [29]. These results suggest that exercise-induced ROS could mediate long-term hormetic effects in WAT, contributing to the overall beneficial effects of exercise. In the present study, we test this hypothesis by biochemically evaluating the effects of chronic aerobic exercise on WAT redox homeostasis and its physiological consequences in adult male Wistar rats.

## 2. Materials and Methods

### 2.1. Experimental Model

Adult male Wistar rats weighing 300–350 g at 10 weeks of age were randomly divided into two groups: the control group (*n* = 8 animals/group) and exercised group (*n* = 8 animals/group). They were allocated in an animal room with controlled temperature (27–29 °C) and lighting (12-h light-dark cycle), and free access to standard rat chow and water (Appendix A). This study was approved by the Institutional Animals Committee for Research (Protocol n°: 132/18), and it has followed the procedures complied with the International Guiding Principles for Biomedical Research Involving Animals of the Council for International Organizations of Medical Sciences (Geneva, Switzerland). 

### 2.2. Chronic Exercise Model

Before beginning the exercise intervention, the animals of the exercised group were adapted to the treadmill (Insight EP 131, Ribeirão Preto, Brazil) for seven days. The adaptation protocol was performed at a speed of 10 m/min for 10 min. After the adaptation, the first maximum speed incremental test was performed. The initial speed was 6 m/min, every 3 min 3 m/min was increased, and the inclination was fixed at 10°. The test was performed until the animal’s exhaustion (when the animal stayed in the steel grid despite increasing shock stimuli) according to previous studies of our group [29,30] (Appendix A).

The maximum reached in the maximal speed test (MST) was used to calculate the speed of the first four weeks of exercise intervention. Exercise periodization was performed with progressive speed, starting with 50% of the maximal speed in the first week, being increased by 5% per week, until the fourth week, and concluding with 65% of maximal capacity. The animals performed the MST at the end of the fifth week to recalculate the training load, preventing the plateau effect due to the adaptations generated by the initial 4 weeks of training. Three days after the second MST, on the sixth week, the exercise group was exposed to the same periodized training protocol previously described, with the new maximum speed baseline, until the ninth week. All exercise sessions lasted 30 min and were then euthanized by decapitation after 72 h on the last day of training (Appendix A). The blood was collected from the neck, retroperitoneal WAT was harvested, and stored at −80 °C until further analyses (maximum of four weeks of storage).

### 2.3. Lactate Measurements 

Blood samples were collected from the tail vein before and immediately after the MST and placed in tubes with 25% sodium fluoride (NaF). To obtain plasma, blood samples were centrifuged at 2000× *g* at room temperature for 15 min. Lactate levels were measured in plasma samples using a commercially available Lactate Bioclin kit (#K084, Quibasa, Brazil), according to manufacturers’ instructions. The samples were analyzed by spectrophotometry at 340 mm in a microplate reader (Victor X4; PerkinElmer). The values were obtained through the product of the sample absorbance by the calibration factor (concentration of the standard curve x standard absorbance), obtaining a concentration in mg/dL. These results were multiplied by × 0.1109 to convert for mmol/dL. The minimum- and maximum sensitivity of the test were 1.0036 mg/dL and 120 mg/dL, respectively, with a coefficient of variation of 0.20% [31,32,33].

### 2.4. Total Body Mass and Adiposity

The animals were weighed in the pre-exercise period as well as in the fourth and the ninth week of training. In the necropsy, adipose tissue compartments were dissected and weighed immediately after euthanasia. The adiposity index was calculated by the sum of the weight of the WAT deposits (epididymal WAT; subcutaneous WAT; retroperitoneal WAT and visceral WAT) as a function of the animal’s final body mass, multiplied by 100 [34].

### 2.5. Histology–Adipocyte Diameter and Number per Area

A sample of the rWAT depot was fixed in 3.7% formaldehyde, processed, embedded in paraffin, and sliced up to 5 µm thick with a microtome. The slides were stained with hematoxylin-eosin for each experimental group, according to a previous protocol [35]. To analyze the adipocyte diameter, random digital images of the rWAT histological sections were captured with a camera attached to the microscope (video microscopy system) with a magnification of 200× [35]. The smallest and largest diameters of at least 50 adipocytes per animal were measured. The Straight-line tool, available in the Software Image J, was used to estimate the mean diameter. The number of fat cells can be determined from the numerical adipocyte density by WAT area (QA). This measure was obtained by counting the number of adipocytes present in a known test area. The test area (TA) was produced using the STEPanizer software (www.stepanizer.com, accessed on 4 November 2021), where QA = N [adipocytes]/TA, a method proposed by Lemonnier [36]. 

### 2.6. NADPH Oxidase Activity

NADPH oxidase activity (NOX) activity was quantified by Amplex Red/Horseradish Peroxidase (HRP) Assay (Molecular Probes, Invitrogen, Waltham, MA, USA). Retroperitoneal WAT homogenization and the NADPH analysis protocols were performed according to the methodology previously described by our group [29,37]. 

### 2.7. Mitochondria Isolation and Measurement of Mitochondrial Function

Immediately after euthanasia of the rats, we performed mitochondria isolation from fresh rWAT. The samples were subjected to differential centrifugation according to the modified protocol of Cladeira et al., (2021) [38] and Maciel et al., (2020) [39]. The isolation of retroperitoneal WAT mitochondria was performed according to the protocol previously published by our group [29,39].

#### 2.7.1. Mitochondrial Oxygen Consumption 

A Clark-type electrode (Strathkelvin, Glasgow, UK) was used to measure mitochondrial respiration. The measurement was performed with a magnetic stirring in a respiration buffer with contained 125 KCl, 10 MOPS, 2 MgCl_2_, 5 KH_2_PO_4_, 0.2 EGTA (mmol·L^−1^) with pyruvate (5 mmol·L^−1^), and malate (5 mmol·L^−1^) for complex I analysis. For complex II analysis, the medium contained succinate (5 mmol·L^−1^) and rotenone (500 µmol·L^−1^). The calibration of the oxygen electrode was performed at a solubility coefficient of 217 nmol O_2_/mL at 37 °C. For the measurement of Complex I respiration, a corresponding 200 µg of mitochondria on 1 mL of incubation buffer was used. A total of 1 mmol·L^−1^ ADP was added after 2 min of incubation, and ADP-stimulated respiration was measured for 3 min. After that, we measured complex IV respiration and maximal uncoupled oxygen uptake. For complex IV respiration analysis it was used N,N,N,N’-tetramethyl-p-phenylenediamine (TMPD, 300 μmol·L^−1^) with ascorbate (3 μmol·L^−1^). Finally, for the maximal uncoupled oxygen uptake measurement, we used 30 nmol·L^−1^ carbonyl cyanide-p-trifluoromethoxyphenyl-hydrazone (FCCP) [40].

#### 2.7.2. Mitochondrial ATP Production

After the oxygen consumption measurement, the mitochondria from the respiration chamber were collected and immediately supplemented with the ATP Assay Mix (diluted 1:5) (Sigma, Aldrich). Mitochondrial ATP production was determined in each sample and compared with ATP standards using a 96-well white plate and a spectrofluorometer (SpectraMax^®^ M3, Molecular Devices, EUA, San Jose, CA, USA) at 560 nm emission wavelength [41].

#### 2.7.3. Mitochondrial ROS Production

Mitochondrial ROS production was measured by the Amplex Red Hydrogen Peroxide Assay Kit (Life Technologies, Carlsbad, CA, USA). The ROS and Amplex Red reaction occur at 1:1 stoichiometry with peroxides under catalysis by HRP, generating highly fluorescent resorufin. The buffer containing mitochondria was supplemented with 50 µmol·L^−1^ Amplex UltraRed and 2 U/mL HRP. After 120 min of incubation in the dark, the supernatant was collected, and mitochondrial ROS concentration was measured and compared with H_2_O_2_ standards using a 96-well plate and a spectrofluorometer (SpectraMax^®^ M3, Molecular Devices, EUA) at 540 nm emission and 580 nm extinction wavelengths [41].

#### 2.7.4. Mitochondrial Swelling and Transmembrane Potential

Mitochondrial swelling and transmembrane potential were analyzed in a spectrofluorometer (SpectraMax^®^ M3, Molecular Devices, USA). Mitochondrial membrane integrity was assessed by osmotically induced changes in mitochondrial volume using the spectrophotometry technique, at an absorption of 540 nm. The mitochondrial suspension (100 µg/mL) was added to the respiratory medium in the absence of respiratory substrates, at 37 °C and under constant agitation. Mitochondrial swelling was stimulated with 100 nmol·L^−1^ calcium, and it was expressed as a percentage of absorption of the solution containing mitochondria in the presence of cyclosporin A (0% mitochondrial turgor), related to that absorbed after the addition of FCCP (100% mitochondrial turgor). To determine the mitochondrial transmembrane potential (∆ᴪm), the TMRM probe (tetramethylrhodamine methyl ester, 400 nmol·L^−1^) was added to an incubation solution containing 100 µg/mL of mitochondria for 1 h. ∆ᴪm was estimated by TMRM fluorescence on 580 nm excitation wavelength. The percentage of fluorescence emitted by TMRM-labeled mitochondria in the presence of cyclosporin A (0% mitochondrial depolarization), with that emitted after the addition of FCCP to fully depolarize the mitochondria (100% mitochondrial depolarization) was used to calculate the ∆ᴪm. To calculate the fraction of electrons leaking from the respiratory chain, the rate of formation of H_2_O_2_ was divided by the rate of mitochondrial O_2_ consumption. 

### 2.8. Antioxidant Enzymes Activities

1 mL of 5 mM Tris-HCl buffer (pH 7.4), containing 0.9% NaCl (*w*/*v*) and 1 mM EDTA was used to homogenize 700 mg of rWAT. The homogenate was centrifugated at 750× *g* for 10 min at 4 °C, we removed the lipid layer and collected the supernatant at −80 °C. The activity of total superoxide dismutase (SOD) was determined by cytochrome C reduction o at 550 nm [42]. The method of Aebi (1984) was used to analyze Catalase (CAT) activity and the results were expressed as units per milligram of protein (U/mg) [43]. To measure the Glutathione peroxidase (GPX) activity we followed NADPH oxidation at 340 nm in the presence of an excess of glutathione reductase, reduced glutathione, and tert-butyl hydroperoxide as substrates. The results were expressed as nmol of oxidized NADPH per milligram of protein (nmol/mg) [44]. Protein concentration was determined by the Bradford assay [45].

### 2.9. Biomarkers of Oxidative Damage

#### 2.9.1. Quantification of GSH and GSSG Levels

We used a colorimetric assay for quantification of oxidized and reduced glutathione (Sigma-Aldrich, USA-Cat. #38185, St. Louis, MO, USA) to determine the ratio of reduced/oxidized glutathione (GSH/GSSG). A total of 100 mg of WAT was homogenized in 700 µL of 5-sulfosalicylic acid to avoid protein interference, as recommended by the manufacturer. The total glutathione content (GSH + GSSG) was assessed first. Then, isolated levels of GSSG were obtained using “reagent masking”, the function of which is to remove GSH. Thus, GSSG in the sample solution was determined by measuring the absorption [λ = 405 nm-from a colorimetric reaction with 5,5-dithionitrobenzoic acid (DTNB)] coupled to the enzymatic recycling system. The amount of GSSG was subtracted from the amount of the total glutathione content to determine the GSG levels [46], and it was expressed in µmol/mg of protein.

#### 2.9.2. Carbonylated Proteins by 2D OxyBlot

The samples were denatured and derivatized with a 12% solution of sodium dodecyl sulfate (SDS) and dinitrophenylhydrazine (DNPH) following the OxyBlot Protein Oxidation Detection Kit manufacturer’s instruction (Millipore–Cat. #S7150). The neutralization solution buffer was used to terminate the derivatization reaction after 15 min. Samples were placed on a 12% bis-acrylamide gel at 120 V for 60–120 min (BioRad) to separate the protein according to their molecular weight, followed by transfer to a nitrocellulose membrane at 25 V overnight. Then, 1x phosphate-buffered saline and Tween 20 (1x PBS-T) and 5% BSA were used to block non-specific binding sites, for one hour. The primary antibody (Anti-Rabbit-NDP-Kit Millipore OxyBlot) was added in a 1:500 dilution, stirred overnight at 4 °C, followed by 3 washes with 1x PBS-T. Then, the membranes were incubated with a goat anti-rabbit IgG (conjugated with peroxidase) antibody (1:300) for one hour at room temperature. The Luminata™ Western HRP (Millipore, Millerica, MA) was used to visualize the protein bands, on BioRad Chemidoc and Image Lab (BioRad, Hercules, CA, USA). For an accurate analysis of protein carbonylation, the samples are presented in duplicates. The first line represents the binding of the protein with the DNP antibody, and the second line, which appears unmarked, is the negative control from the same sample that was exposed to the same conditions but in the absence of DNP. The objective of presenting it in this way is to prove through the negative control that the observed signal is specific to the binding of DNP, according to the manufacturer’s instruction. We normalized the densitometric values of the dinitrophenylhydrazone (DNP) bands by the endogenous control (β-actin) and then normalized it to the control group [47].

### 2.10. Protein Expression Analysis–Western Blotting

Retroperitoneal WAT (700 mg) was homogenized in 1 mL of 5 mM Tris-HCl buffer (pH 7.4), containing 0.9% NaCl (*w*/*v*) and 1 mM EDTA, centrifugated at 750× *g* for 10 min at 4 °C. A total of 50 µg of protein homogenate was subjected to denaturing polyacrylamide gel electrophoresis using a linear gradient of 8–12% (SDS-PAGE), according to the molecular weight of each target protein. Then, the transfer of the gel samples to PVDF membranes was performed. Membranes were stained with 0.2% Ponceau’s red solution to confirm the transfer of protein bands and subsequently washed with 0.1% TBS-Tween. In sequence, the membranes were blocked in 0.1% TBS-Tween buffer containing 5% BSA for 1 h at room temperature and incubated overnight at 4 ºC with the determined primary antibodies (Total OXPHOS Rodent WB Antibody Cocktail, Abcam, #ab110413 1:1000; 4-HNE, Abcam, #MA5-27570 1:700; H2AX, Cell Signaling #2595 1:1000; γH2AX, Cell Signaling #2577 1:1000; NF-Κb p65, Invitrogen #51-0500 1:250; p-NF-Κb p65, Invitrogen #MA5-15160 1: 1000; Nrf2, Invitrogen #PA5-88084 1:1000; p-Nrf2, Invitrogen #PA5-102838 1:1000; p53, Cell Signaling #2524 1:1000; p-P53, Cell Signaling #9287 1:1000; p21, Santa Cruz #SC- 397 1:500; p-p21, Abcam # ab47300 1:700, and β-actin, Cell Signaling #4967 1:1000). The antigen-antibody complex was detected with the secondary antibody conjugated to HRP (specific for each primary antibody) and the identification of proteins was performed by comparing the bands obtained in the immunoblot with the bands related to the protein standard submitted to the same methodology as the samples (β-actin). Protein expression quantification was performed by densitometric analysis of the intensities of the bands obtained, analyzed in the ImageJ program of the National Institute for Health (USA) (Bethesda, MA, USA).

### 2.11. Cytokine Profile

150 mg of rWAT was homogenized in 5 mM Tris-HCl buffer (pH 7.4), containing 0.9% NaCl and 1 mM EDTA, followed by centrifugation at 750× *g* for 10 min at 4 °C. Commercial kits (DuoSet^®^) were used to analyze the supernatant pro-inflammatory cytokines. The rat tumor necrosis factor kit (TNF-α; Cat. DY510-05) and the rat interleukin-6 (IL-6; #Cat. DY506-05) were performed using an enzyme-linked immunosorbent assay (ELISA) (R&D Systems Inc. kits-Minneapolis, MN, USA) with a detection range of 62.5–4000 pg/mL for TNF-α, and 125.0–8000 pg/mL for IL -6 concentrations. All reagents were prepared in advance according to the manufacturer’s instructions and kept at room temperature. A microplate reader (Victor X4; PerkinElmer, Norwalk, CT, USA) was used to read the optical density at 450 nm with a correction of 570 nm. Results were expressed in p/mg of protein.

### 2.12. Gene Expression by Real-Time qPCR

The tissue disruption was performed by a Dynabeads™ and TissueLyser LT to extract total RNA from rWAT. RNeasy Lipid Tissue Mini Kit (Qiagen, Hilden, Germany) was used for RNA extraction, according to the manufacturer’s instructions. To determine the RNA concentration and purity it was measured the sample’s absorbance at 260 and 280 nm with a spectrophotometer (Biomate 3S, Thermo Scientific, Waltham, MA, USA), and integrity was analyzed by electrophoresis in agarose gel (1%). The High-Capacity cDNA Reverse Transcription Kit (Invitrogen, Waltham, MA, USA) was used to synthesize the cDNA, and it was used 1.4 µg of RNA in a thermocycler (Techne TC-412, Cambridge, UK) according to the manufacturer’s instructions. SYBR Green (Thermo Scientific) was used to perform the real-time PCR. The qPCR cycle was set according to the manufacturer’s instructions (initial denaturation 95 °C for 15 min once; followed by denaturation 95 °C for 15 s plus annealing 60–65 °C for 20 s and elongation 72 °C for 20 s, repeated 40 times). The housekeeping control gene used was the ACTB gene. Finally, gene expressions were analyzed using the 2^−ΔΔCt^ method [48]. Sequences of primers used for real-time PCR are referred on Appendix A. 

### 2.13. Statistical Analysis

The results were expressed as mean ± standard error of the mean (SEM) and analyzed through the statistical program GraphPad Prism 7.0 (San Diego, CA, USA). D’Agostino and Pearson test was used to verify the normal distribution of the data, followed by Paired T-test for the lactate and performance measurements. The Unpaired *t*-test was used for the other analysis. The minimum significance level established was *p* < 0.05.

## 3. Results

### 3.1. Aerobic Exercise Performance 

To test our hypothesis that exercise-induced ROS could mediate long-term hormetic effects in WAT we employed a model of chronic aerobic exercise in adult male Wistar rats. We validated our exercise regimen by measuring plasma lactate levels, which were higher after the maximal exercise test compared to resting conditions in the first and second tests (Appendix A). These results show that all animals reached a very high effort level, as plasma lactate was higher than 7 mmol/L, which ensures greater accuracy in the calculation of the % of the maximum individual running speed. In the second bout, the running speed, the distance covered, and the duration of the test were higher compared to before (Appendix A). These results indicate an adaptative response to the exercise program. Moreover, we observed that the maximal speed and the total running distance in the eighth week were higher for each rat compared to the performance in the fourth week, with the same absolute intensity (65%)—(Appendix A).

### 3.2. Effects of Chronic Aerobic Exercise on Body Weight, Adiposity, and WAT

As expected, we observed that our exercise regimen reduced body weight after eight weeks of exercise, compared to the fourth week of exercise as well as compared to the sedentary group (Figure 1a). This was associated with lower total WAT weight and lower retroperitoneal, gonadal, and subcutaneous WAT depot weights in the exercise group in comparison to the controls, which resulted in a lower adiposity index (Figure 1b–d). Analyzing the histology of WAT (Figure 1e) we found that the number of adipocytes per area was higher and the diameter of adipocytes was lower in comparison to the CTRL group (Figure 1f,g). Moreover, mRNA expression levels of lipogenesis-related genes Vegf, Pparg, and Srebf1 were lower in the exercise group (Figure 1h).

### 3.3. Effect of Chronic Aerobic Exercise on Mitochondrial Function in WAT

As structural changes in adipose tissue are usually accompanied by functional changes in energy metabolism [49], we assessed the effect of exercise on mitochondrial function. Interestingly, we observed that mitochondria isolated from WAT of the exercise group displayed lower mitochondrial oxygen consumption, in state I of Complex I. However, mitochondrial O_2_ consumption in states II, and III of Complex I was higher in WAT from exercised rats compared to the sedentary group. Regarding mitochondrial complex IV, respiration and maximal oxygen uptake of uncoupled mitochondria were not different between groups, indicating an equal loading of viable mitochondria in all experiments (Figure 2a).

Regarding Complex II, mitochondrial O_2_ consumption in state I was higher, and a tendency of increase in state II was observed in the WAT of the exercise group compared to controls, without significant alterations in mitochondrial complex IV respiration and maximal oxygen uptake of uncoupled mitochondria between groups (Figure 2b). In addition, we observed that ∆ᴪm was more negative and the ATP production higher in the exercise group compared to the control (Figure 2c,d). Interestingly, no differences between groups were found for mitochondrial ROS production (Figure 2e). However, the ROS/ATP ratio was lower in mitochondria from the exercise compared to the sedentary group (Figure 2f), and no differences in electron leakage were found between groups (Figure 2g). Taken together, these results demonstrate an increase in mitochondrial efficiency in mitochondria isolated from WAT of chronically exercised rats.

In addition to the mitochondrial respiratory enhancement, we found higher protein levels of mitochondrial complexes in WAT from the exercise group compared to controls (Figure 3a–f). This effect was seen in CII, CIII, and CV, but not in CI and CIV levels (Figure 3a–f). These results demonstrate that chronic exercise enhanced mitochondrial abundance, the efficiency of mitochondrial ATP production, and mitochondrial respiration.

### 3.4. Effect of Chronic Aerobic Exercise Exposure on Redox Homeostasis and Oxidative Biomarkers

Since no changes in mitochondrial ROS production were observed, we evaluated other parameters of redox homeostasis. We found lower NOX activity, and higher SOD, GPx, and Catalase activities (Figure 4a–d) in the WAT isolated from the exercise group in comparison to the sedentary group. Altogether, these changes indicate an increase in the antioxidant capacity associated with a reduction in ROS production.

Given the increase in the antioxidant capacity in WAT after exercise, we evaluated the capacity of exercise to activate the NRF2 pathway, the master transcriptional regulator of the antioxidant response. Interestingly, our results demonstrate that the exercise group displayed higher levels of total NRF2 protein than the sedentary group (Figure 5a). Nevertheless, this was not associated with significant changes in the gene expression of antioxidant proteins (Figure 5c).

As expected, the enhanced antioxidant capacity in the exercise group was followed by lower levels of oxidative damage markers. GSSG levels were lower and GSH levels were higher, resulting in a higher GSH/GSSG ratio (Figure 6a–c). Moreover, lipid peroxidation (Figure 6d) and protein carbonyl levels (Figure 6e) were also lower in the exercise group compared to the control group. These results show that exercise was associated with lower markers of oxidative damage in WAT compared to sedentary animals.

### 3.5. Chronic Aerobic Exercise Exposure Does Not Change Inflammatory Markers, but Increases IL-6 Gene Expression on WAT-r

It is well known that NF-κB is one of the critical key nodes for integrating inflammation, stress, and metabolism [50]. As we observed that exercise led to lower ROS production, we evaluated the activation of NFκB and the levels of inflammatory cytokines in WAT. We did not find any difference between the groups for NFkB-p65 activation (Figure 7a). However, we observed that WAT from the exercise group showed a tendency to lower TNF-α levels, and no differences were found in IL-6 levels compared to the control group (Figure 7b,c). Interestingly, exercise increased *Il6* mRNA levels by 5-fold in comparison to sedentary rats, without any differences in Tnf and Ccl2 mRNA levels (Figure 7d). 

### 3.6. Chronic Aerobic Exercise Exposure Decreases DNA Damage in WAT

Improvements in antioxidant defense are generally related to lower DNA damage [51,52]. To determine whether exercise-induced changes in mitochondrial function and redox homeostasis are linked to enhanced DNA protection, we evaluated DNA damage response markers in our experimental groups. We observed that γH2AX was significantly lower in the exercise group compared to controls (Figure 8a,b). However, no differences were found in total H2AX protein expression, p21 and p53 activation (Figure 8a,c–e). Curiously, these results were not associated with changes in cellular senescence- and cell cycle arrest-related genes. Cdkn2a and Cdkn2d mRNA levels were not different between groups (Figure 8g). 

## 4. Discussion

It is well established that exercise is an important component of improving metabolic health [53]. Several pieces of evidence point to the beneficial effects of exercise in WAT, as exercise increases lipolysis and mitochondrial function as well as improves insulin sensitivity [54]. In the present study, we showed that 9 weeks of aerobic exercise induced an adipose tissue phenotype protective against oxidative stress in WAT. Our findings demonstrated that the exercise-induced adipose tissue remodeling was linked to increased mitochondrial function and enhanced redox homeostasis. From the perspectives of adipose tissue biology, these findings demonstrate that exercise is an important intervention capable of improving the metabolic capacity, remodeling its structure, and increasing the defense against oxidizing agents in adipose tissue. 

Obesity is a condition that is usually associated with oxidative stress in adipose tissue. Compared to other depots, retroperitoneal WAT shows larger adipocytes, higher amounts of lipogenic transcription factors, and low expression of fatty acid oxidation genes, which makes it more susceptible to metabolic and energetic challenges [55,56]. We found that our exercise protocol effectively lowered overall adiposity, which was also reflected in lower adipocyte diameter in WAT. The morphological alterations were accompanied by a reduction in the expression of genes related to lipogenesis. These results agree with the literature, which points to a reduction in adiposity and lipogenesis by exercise [57,58]. However, WAT remodeling is linked to physical capacity independently of weight loss (Appendix A). De Melo et al. (2022) also demonstrated these results in the resistance training model, which aimed to investigate the influence of short-term strength training on lipolysis, lipogenesis, and browning processes in the subcutaneous adipose tissue in obese mice. It was demonstrated that strength training increased browning process-related genes in the subcutaneous adipose tissue, independently of changes in body weight [59]. Exercise induces metabolic improvements by the enhancement of mitochondrial respiration and ATP production in several tissues, including WAT [60]. However, no evidence was found demonstrating the role of exercise in the relationship between mitochondrial function and redox homeostasis in WAT. Most studies point to a pathophysiological context associated with a sedentary lifestyle. In our previous study, we demonstrated that one session of aerobic exercise increased respiratory activity, ATP, and ROS production in a transient way in mitochondria isolated from retroperitoneal WAT [29]. Although one acute exercise bout acts as a stressor to redox- and bioenergetic homeostasis, the literature is clear in demonstrating the benefit of chronic exposure to physical exercise on mitochondrial function [60,61,62,63]. In the present study, chronic aerobic exercise increased mitochondrial respiration, resulting in a higher flux of protons to the intermembrane space, which favors a greater transmembrane potential and a higher driving force for ATP synthase activity. These results demonstrate that exercise provides an adaptive mitochondrial response to produce ATP more efficiently, due to the more negative ∆ᴪm, which was associated with a lower ROS/ATP ratio. Furthermore, among the changes induced by exercise, the increase in intracellular calcium concentration, ROS production, and the AMP/ATP ratio are important mediators for PGC-1α and Transcription factor A mitochondrial (TFAM) activation. These transcription factors play an important role in maintaining mitochondrial integrity, function, and morphology [64].

In addition, exercise increased the expression of CII, CIII (a complex responsible for shunting protons across the intermembrane space and bringing electrons to complex IV) [65], and CV (responsible for ATP production). These results indicate that exercise increases mitochondrial respiratory chain protein expression and are in agreement with the results of mitochondrial function (increase in respiration and ATP production). Enhanced mitochondrial efficiency can be directly related to the increase in ATP production but producing similar ROS levels [66]. These changes are directly related to improvement in metabolic processes of several tissues [67,68]. Regarding WAT, they are related to increasing thermogenesis capacity and induce a browning phenotype [69]. 

Exercise-induced mitochondrial efficiency is directly associated with the modulation of redox homeostasis [70]. It is known that under deleterious conditions, such as obesity, aging, and a sedentary lifestyle, mitochondria not only show loss of function and increased production of ROS, but NOX activity is also increased [71]. NOX enzymes are a family of transmembrane proteins that transport electrons across biological membranes, thereby reducing O_2_ to O_2_^●−^, as well as to H_2_O_2_ [72,73,74,75]. These enzymes play an important role in redox-sensitive reactions and cell signaling. However, in deleterious conditions NOX activities are increased, resulting in higher ROS production, which can lead to oxidative damage in biomolecules [76]. Among the different NOXs isoforms, Nox2 and Nox4 are the most studied in the adipose tissue physiological context. These isoforms are crucial to adipocyte differentiation, and metabolic- and proteostatic homeostatic control of the adipocyte [77,78,79]. In our study, the exercise did not change Nox 2 and 4 mRNA expression but decreased NOX activity followed by enhanced activities of the three main antioxidant enzymes SOD, CAT, and GPx in WAT. It has been shown that the antioxidant activity is positively modulated by the NRF2 pathway, being directly related to hormesis [80]. In the present study, long-term exercise increased NRF2 expression, which was probably related to a hormetic effect of exercise. Previously, we demonstrated that an acute bout of exercise induced a transient pro-oxidative state in WAT, which was linked to higher *Nfe2l2* gene expression [29]. Thus, we suggest that an increase in ROS availability stimulated by each acute exercise session provokes Keap1 oxidation, reducing NRF2-Keap1 interaction and, consequently, increasing NRF2-mediated transcriptional activity of genes containing antioxidant response elements (AREs), resulting in the transcription of more than 200 cytoprotective genes, including NRF2 itself [81,82]. Several studies demonstrate that NRF2 expression and activation increase in different tissues, such as muscle, brain, and heart, after a training period above 6 weeks in different tissues [83,84,85,86], but until now NRF2 in WAT remained unclear. We did not find any difference in antioxidants-related genes and NRF2 phosphorylation and antioxidants-related gene expression. We attribute these results to the time of tissue collection, as the animals were euthanized 72 h after the last training session. According to Egan, B. and Zierath, J. R., (2013), a single exercise bout elicits a rapid, but transient, increase in relative mRNA expression. Alterations in mRNA expression from basal levels are typically observed 3–12 h after cessation of the exercise and generally return to the basal levels within 24 h [87]. However, we observed an increase in the total expression of the NRF2 protein. As NRF2 is subject to posttranslational regulation and its protein levels are higher in the exercise group, we attribute that this pathway was previously activated by exercise sessions [15].

The reduction in ROS production and the improvement of antioxidant defense can directly impact oxidation damage reactions that occur within the tissue. We found a three-fold increase in the GSH/GSSG ratio. GSH is an important non-enzymatic antioxidant compound that also participates in the chemical reaction of antioxidant enzymes, such as GPx [88]. Increases in GSH/GSSG levels indicate enhanced defense capacity, as well as a reduction in the oxidation of biomolecules, and this is reflected by the lower 4-HNE levels found in our study. 4-HNE is a product of lipid peroxidation, being one of the most reactive aldehydes [89]. It participates in multiple physiological processes as a nonclassical secondary messenger. Moreover, it forms adducts with cysteine residues of thiol-containing proteins. Its deleterious effects are frequently found to be proportional to the number of adducts formed affecting negatively mitochondrial ATPase activity, oxygen consumption, and even increasing protein damage [90]. Interestingly, our results showed lower protein oxidation levels in the exercised group, pointing to a protective and preventive role of exercise in oxidative damage to lipids and proteins. An important consequence of oxidative stress is the activation of the inflammatory response. It is known that cytosolic H_2_O_2_ induces the phosphorylation of the p65 subunit at Ser-536 leading to NF-kB activation [91], and in line with lower ROS after exercise, we observed that exercise did not alter the phosphorylation of NFkB-p65. Furthermore, we found a tendency to reduce in TNF-α levels and an increase in *Il6* gene expression in the exercised group. IL-6 has a dual role and can act as an anti- or pro-inflammatory cytokine [92,93]. Under inflammatory conditions, such as obesity and aging, the increase in IL-6 levels is accompanied by higher levels of TNF-α and its receptors, which are responsible for inducing pro-apoptotic responses [94]. However, in response to exercise, the increase in IL-6, concomitant to the increase in several anti-inflammatory adipokines, exerts anti-inflammatory effects, induces lipolysis on WAT, and improves insulin sensitivity [95]. Although we did not find an increase in IL6 levels, it is interesting to underline its increase in gene expression. 

Oxidative stress in WAT can induce strand breaks, base changes, and abasic site formation, activating the DNA damage response pathway, which is related to the impairment of adipocyte function in obesity and aging [96]. Histone H2AX phosphorylation is increased by DNA damage and plays a key role in the DNA damage response [97], which is related to cellular cycle arrest, leading to senescence phenotype and apoptosis [98]. We demonstrated that exercise was associated with lower γH2AX levels, potentially indicating enhanced DNA protection, which could be a prophylactic strategy to preserve the physiological function of adipose tissue through exercise in aging.

Nevertheless, it is important to highlight the limitations of the present study. An important point is the lack of comparison between different compartments of WAT and other organs. It could contribute to a better understanding of the specific responses of different WAT deposits and tissues to exercise training. Moreover, our measurements were made 72 h after the last training session to observe the chronic role of exercise without interference from the last session. Analysis in shorter times after the last exercise session (24 and 48 h) could better clarify the activation of the NRF2 pathway in our model. Finally, although we demonstrated an increase in antioxidant defense and a decrease in oxidative and DNA damage markers after chronic exercise, experiments demonstrating a direct relationship between them were not performed, such as the evaluation of these responses in an exercised group treated with an antioxidant. Furthermore, it is important to note that this effect was observed in eutrophic, not obese, animals. Demonstrating the prophylactic role of aerobic exercise in this model and opening new gaps in the literature for a possible continuation of this work in metabolic and inflammatory conditions, such as obesity.

## 5. Conclusions

In conclusion, our work defines the antioxidant effects of chronic aerobic exercise in adipose tissue, decreasing ROS production and preventing oxidative damage to proteins and DNA. These adaptive responses were associated with the activation of NRF2 and enhanced mitochondrial efficiency. Our study suggests a potential hormetic role of chronic exercise that may protect or enhance adipose tissue function in obesity and ageing.

## Figures and Tables

**Figure 1 antioxidants-11-01689-f001:**
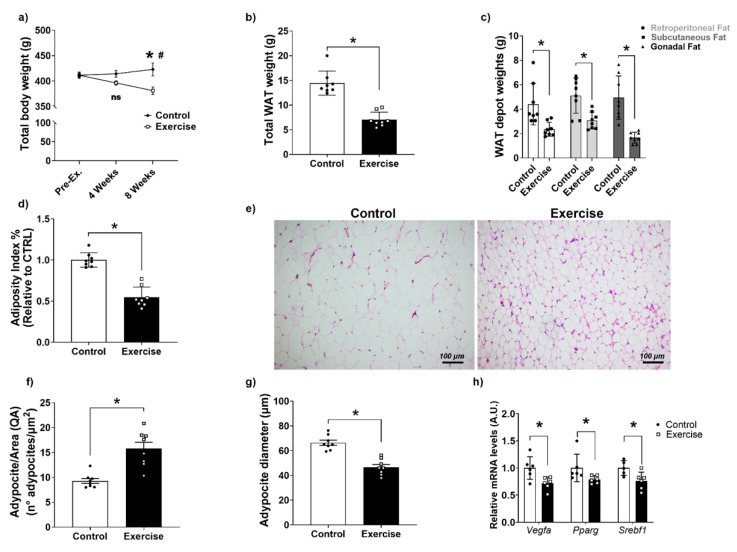
Aerobic exercise reduced body weight and adiposity in Wistar rats. (**a**) Body weight; (**b**) Total WAT weight; (**c**) Weight of WAT depots; (**d**) Adiposity index; (**e**) WAT hematoxylin-eosin histology; stained with hematoxylin-eosin at 200x magnification; (**f**) Histomorphometry of numerical adipocyte density (QA); (**g**) Histomorphometry of adipocyte diameter; (**h**) mRNA levels of genes related to lipogenesis in WAT. Pparg: Peroxisome proliferator-activated receptor-gamma; Srebf1: Sterol regulatory element-binding protein 1; Vegf: Vascular Endothelial Growth Factor. The data were expressed as the mean ± standard error of the mean (*n* = 8 animals/group). •—Control group; **▫**—Exercise group. ^#^ Statistical difference intra-group related to the fourth week, and * Statistical difference between groups.

**Figure 2 antioxidants-11-01689-f002:**
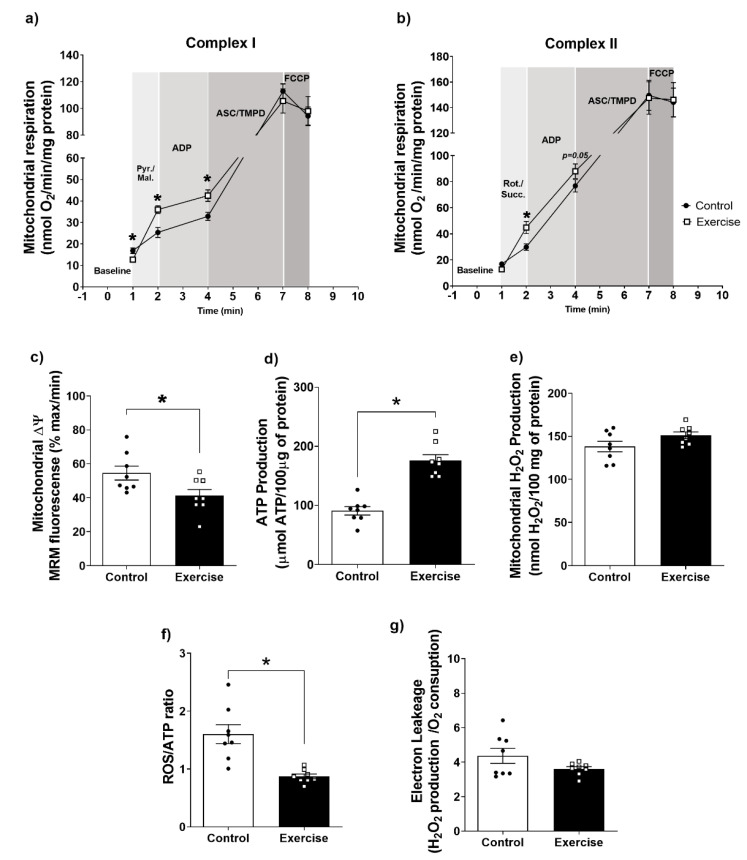
Effects of aerobic exercise on mitochondrial function in retroperitoneal WAT. Analysis of complex I (**a**) and complex II (**b**) mitochondrial respiration; (**c**) Transmembrane potential (∆ᴪm); (**d**) Mitochondrial ATP production; (**e**) Mitochondrial ROS production (H_2_O_2_); (**f**) ROS/ATP ratio; (**g**) Electron leakage. The data were expressed as the mean ± standard error of the mean (*n* = 8 animals/group). •—Control group; **▫**—Exercise group. * Statistical difference between groups.

**Figure 3 antioxidants-11-01689-f003:**
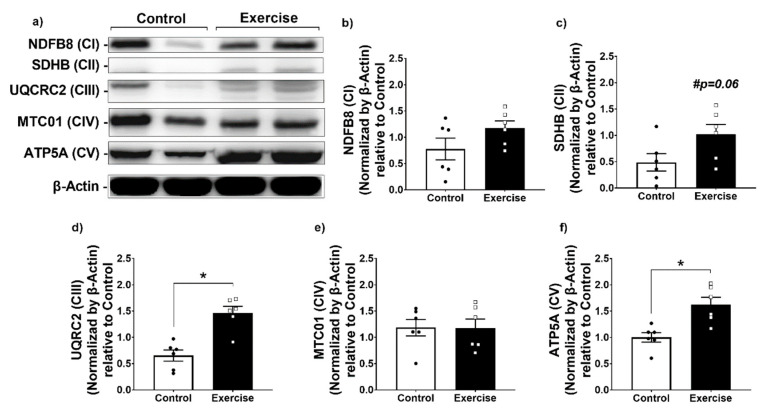
Aerobic exercise increased mitochondrial complexes expression in retroperitoneal WAT. (**a**) Representative image of the expression of mitochondrial complexes by Western blotting; (**b**) NDFFB8 (CI); (**c**) SDHB (CII); (**d**) UQCRC2 (CIII); (**e**) MTC01 (CIV), and (**f**) ATP5A (CV) expressions. The data were expressed as the mean ± standard error of the mean (*n* = 6 animals/group). •—Control group; **▫**—Exercise group. # Statistical tendency = *p* < 0.06. * Statistical difference between groups.

**Figure 4 antioxidants-11-01689-f004:**
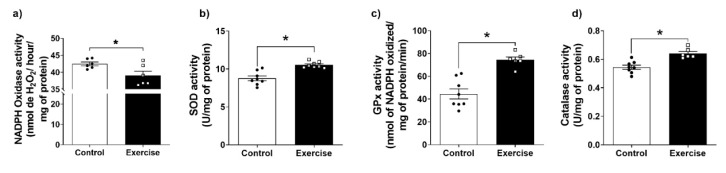
Aerobic exercise improved the antioxidant system in WAT. (**a**) NADPH oxidase activity; (**b**) Total superoxide dismutase (SOD)-; (**c**) Total catalase (CAT)- and (**d**) Total glutathione peroxidase (GPX) activities. The data were expressed as the mean ± standard error of the mean (*n* = 8 animals/control and *n* = 8 animals/group). •—Control group; **▫**—Exercise group. * Statistical difference between groups.

**Figure 5 antioxidants-11-01689-f005:**
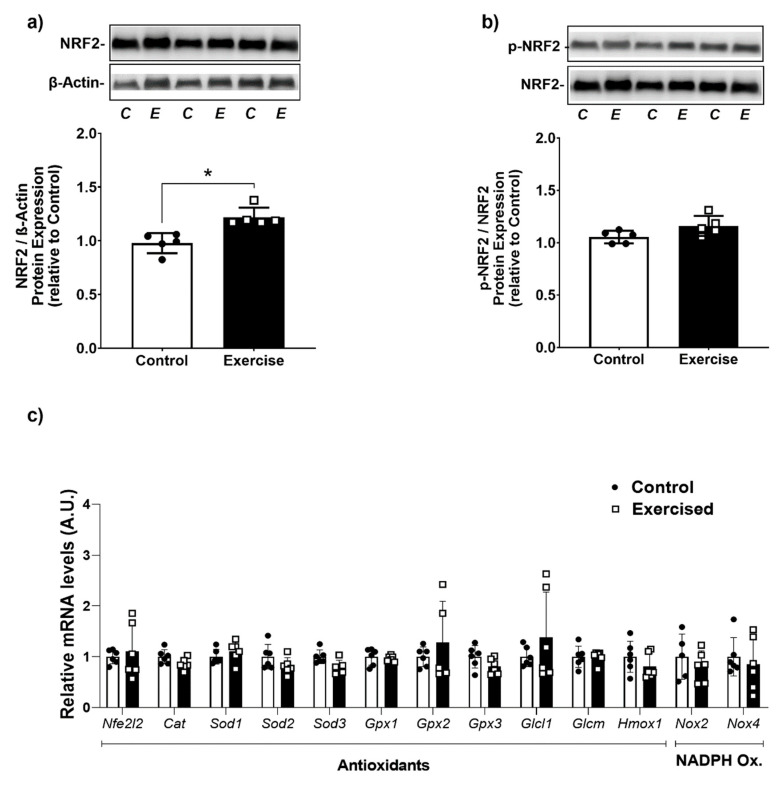
NRF2 was increased by aerobic exercise in retroperitoneal WAT. Total (**a**) and phosphorylated levels (**b**) of NRF2 protein-(*n* = 5 animals/group); (**c**) mRNA levels of pro and antioxidant proteins. Nfe2l2: Nuclear factor erythroid 2-related factor 2; Cat: Catalase: Sod1-3: Superoxide dismutase 1-3; Gpx1-3: Glutathione peroxidase 1-3; Gclc1: Catalytic Subunit 1 Glutamate-Cysteine Ligase; Gclm: Glutamate-cysteine ligase modifying subunits; Hmox1: Heme Oxygenase 1. Nox2 and 4: NADPH Oxidase 2 and 4. The data were expressed as the mean ± standard error of the mean (*n* = 6 animals/group). •—Control group; **▫**—Exercise group. * Statistical difference between groups.

**Figure 6 antioxidants-11-01689-f006:**
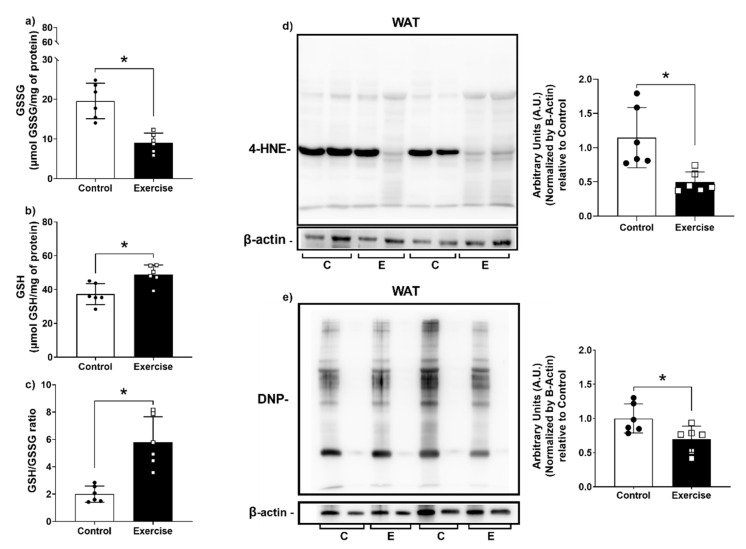
Aerobic exercise reduced oxidative damage markers in WAT. (**a**) Content of GSSG; (**b**) Content of GSH; (**c**) GSH/GSSG ratio; (**d**) Levels of lipid peroxidation; (**e**) Levels of carbonylated proteins. GSSG: Glutathione disulfide (oxidized); GSH: Reduced Glutathione; 4-HNE: 4-Hydroxynonenal; DNP: 2,4-Dinitrophenyl. The data were expressed as the mean ± standard error of the mean. (*n* = 6 animals/group). •—Control group; **▫**—Exercise group. * Statistical difference between groups.

**Figure 7 antioxidants-11-01689-f007:**
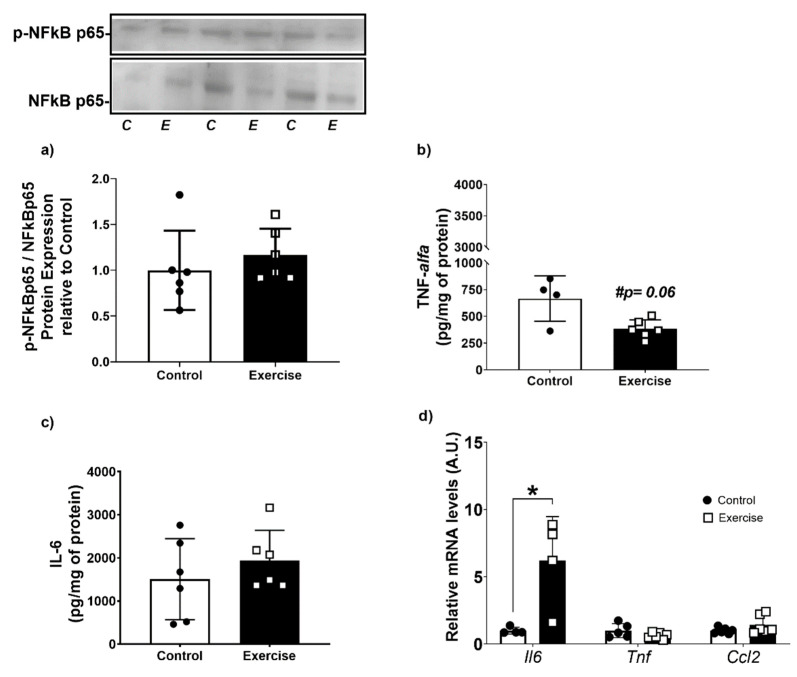
Aerobic exercise did not change inflammatory markers in WAT. (**a**) Nfkb-p65 phosphorylation; (**b**) TNF-alfa cytokine levels; (**c**) IL-6 cytokine levels; (**d**) mRNA expression of genes related to inflammation; NFkB-p65: Factor nuclear kappa B–subunit 65; Il6: Interleukin 6; Tnf: Tumors necrosis factor-alpha; Ccl2: CC motif chemokine ligand 2; The data were expressed as the mean ± standard error of the mean (*n* = 6 animals/group). •—Control group; **▫**—Exercise group. # Statistical tendency = *p* < 0.06. * Statistical difference between groups.

**Figure 8 antioxidants-11-01689-f008:**
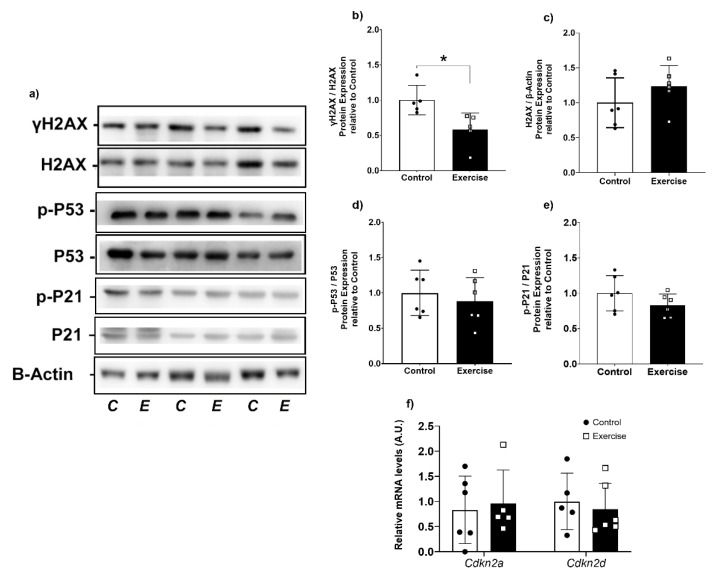
Aerobic exercise reduced the DNA damage response pathway in retroperitoneal WAT. (**a**) Western Blotting representation of proteins related to DNA damage response pathway (H2AX, p53, p21); (**b**) γH2AX phosphorylation; (**c**) H2AX/B-Actin expression; (**d**) p-P53 phosphorylation; (**e**) p-P21 phosphorylation; (**f**) Gene expression related to Cell cycle arrest. All western blot data were normalized to total protein or *β-actin* and expressed as relative to control. γH2AX: phosphorylated- H2A histone family member X; H2AX: H2A histone family member X; Cdkn2a: cyclin-dependent kinase inhibitor 2A; Cdkn2d: cyclin-dependent kinase inhibitor 2D. The data were expressed as the mean ± standard error of the mean (*n* = 6 animals/group). •—Control group; **▫**—Exercise group. * Statistical difference between groups.

## Data Availability

Data is contained within the article and Appendix A.

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
