# Peer review of "Exercise Improves Redox Homeostasis and Mitochondrial Function in White Adipose Tissue"

_antioxidants, 2022, doi:10.3390/antiox11091689_

Round 1

Reviewer 1 Report

Matta et al. propose the work "CHRONIC AEROBIC EXERCISE STIMULATES NRF2, RE- 2 DOX HOMEOSTASIS, AND MITOCHONDRIAL FUNCTION 3 IN WHITE ADIPOSE TISSUE".

The work is well designed.

To improve the quality/presentation of the manuscript/data, I recommend the author address the following points/questions:

1) Review the text to polish typos and make more appropriate English sentences.

2) The authors used rats at 10 weeks of age (300-350g). The authors mention (introduction and discussion) a previous work (published in 2021 by the same research group - ref. 25), in which the rats were 18 weeks of age (400-450 g). I suggest the authors discuss the meaning of age in their protocol designs and results.

3) I strongly suggest that the authors include considerations about the potential limitations (including limitations related to the measurements (techniques) of oxidative/antioxidative biomarkers).

4) The study is founded on the aerobic exercises affecting WAT. I suggest considering the importance of including comparisons with resistance training (literature search) vs. the results the author found under aerobic conditions. This would enrich the article discussion and the readers' interest. Particularly discussing the oxidative/antioxidative processes triggered (modulated) by resistance with the current data under aerobic stimuli.

5) Fig. 6b: is there (or not) a difference between control and exercise groups? (lines 420-421).

Author Response

Responses to reviewer #1

We thank the reviewer for her/his critical feedback.

1) Review the text to polish typos and make more appropriate English sentences.

Answer: Thank you for critically reading our manuscript, we have revised the text and sought to make the English language more appropriate.

2) The authors used rats at 10 weeks of age (300-350g). The authors mention (introduction and discussion) a previous work (published in 2021 by the same research group - ref. 25), in which the rats were 18 weeks of age (400-450 g). I suggest the authors discuss the meaning of age in their protocol designs and results.

Answer: Thank you for this comment. 10 weeks of age was when treadmill running training began. The animals started the intervention protocol at 10 weeks and there were 9 weeks of training. They were euthanized at 19 weeks, with a difference of only 1 week of age for the animals of the previous study.

3) I strongly suggest that the authors include considerations about the potential limitations (including limitations related to the measurements (techniques) of oxidative/antioxidative biomarkers).

Answer: Thanks for this suggestion. Undoubtedly, it is extremely important to recognize the limitations of our study and raise new hypotheses. As requested, we have inserted at the end of the discussion, before the conclusion, a paragraph about the limitations and prospects of our study (lines: 708-722). It is described below:

“Nevertheless, it is important to highlight the limitations of the present study. An important point is the lack of comparison between different compartments of WAT and other organs. It could contribute to a better understanding of the specific responses of different WAT deposits and tissues to exercise training. Moreover, our measurements were made 72 hours after the last training session to observe the chronic role of exercise without interference from the last session. Analysis in shorter times after the last exercise session (24 and 48 hours) could better clarify the activation of the NRF2 pathway in our model. Finally, although we demonstrated an increase in antioxidant defense and a decrease in oxidative and DNA damage markers after chronic exercise, experiments demonstrating a direct relationship between them were not performed, such as the evaluation of these responses in an exercised group treated with an antioxidant. Furthermore, it is important to note that this effect was observed in eutrophic, not obese, animals. Demonstrating the prophylactic role of aerobic exercise in this model and opening new gaps in the literature for a possible continuation of this work in metabolic and inflammatory conditions, such as obesity.

4) The study is founded on the aerobic exercises affecting WAT. I suggest considering the importance of including comparisons with resistance training (literature search) vs. the results the author found under aerobic conditions. This would enrich the article discussion and the readers' interest. Particularly discussing the oxidative/antioxidative processes triggered (modulated) by resistance with the current data under aerobic stimuli.

Answer: Thank you for your detailed comments regarding comparing the effect of aerobic and strength exercise. As requested, we have added this topic to our discussion. In lines (589-595) a comparison between the adaptive responses of aerobic and strength exercise about adipose tissue remodeling was described, below:

Lines (572-578): "However, WAT remodeling is linked to physical capacity independently of weight loss (Supplementary Figure 1d-1f). De Melo et al. (2022) also demonstrated these results in the resistance training model, which aimed to investigate the influence of short-term strength training on lipolysis, lipogenesis, and browning processes in the subcutaneous adipose tissue in obese mice. It was demonstrated that strength training increased browning process-related genes in the subcutaneous adipose tissue, independently of changes in body weight [61].”

5) Fig. 6b: is there (or not) a difference between control and exercise groups? (lines 420-421).

Answer: Very well observed, thank you very much. It was fixed. There is not a difference between the Ex- and control groups on the 6b figure. Only in Figure 6a.

Reviewer 2 Report

Leonardo Matta et al investigate how chronic exercise acts on adipose tissue, with a specific focus on mitochondrial bioenergetics and antioxidant activities. They conclude that 

 chronic effects of physical exercise ameliorate these aspects in retroperitoneal WAT, together with increasing NRF2 expression and reducing DNA damage.

The topic is interesting, however many conclusions are not strongly supported by data. I do not suggest further consideration of the manuscript in this journal.

·      It is not clear in the abstract why γH2AX and p21 investigation is important.

·      Please define “hormetic” the first time the term is mentioned.

·      Provide molecular mechanisms by which improves Δᴪm, -respiration, -fatty acid oxidation, -structural and -functional integrity of the matrix, and -ATP synthesis.

·      It would be important that authors mentioned or reported any literature about chronic exercise that actually evolves being too much and provokes major oxidative stress.

·      A major aspect of concerns regards the utilization of  only retroperitoneal WAT for their analyses. Although authors briefly explain its importance in the discussion, it will be necessary that authors assess at least some of the parameters studied, in other WATs, especially visceral, since its importance for metabolic health is apparent. Authors could also consider studying some bioenergetics in the liver as well.

·      Authors should look at total cellular ROS as well.

·      Figure 2A-B is somewhat unclear. Why this says pyr/mal while in the methods it says glutamate? Moreover, what is the difference between baseline medium between CI and CII? Are the substrates given at baseline? If so, authors should put the nomenclature differently (they should all be shifted) to the previous point of injection on the left. Finally, authors should explain why CI baseline is significantly lower during exercise. What are the major sources for mitochondrial activity in adipose tissue? Does exercise change them? Please cite literature on this regard.

·      The n2 control in the blots of fig3 is confusing. Authors should show more blots. it is really hard to appreciate differences if the second sample is just so low.

·      SDHB this is barely seen in Fig3a

·      Authors should do native blots of mitochondrial complexes. As the western blots stand so far in the current manuscript, considering the confusing levels in the control, data do not seem very compelling and strong.

·      Fig 5e is confusing. Authors should better specify samples for each lane. Yhy they look so different alternatively. Can you expose more?

·      It would appear more appropriate that authors put the NRF2 part before fig 5.

·      Authors should explain why the increase in NRF2 protein does not translate in high gene expression of antioxidant genes? Authors should do western blot on all those proteins or staining in the tissues. They should also interrogate nuclear and cytosolic NRF2.

·      Authors should be careful when stating that they observe lower ROS production, since they show that mitochondrial ROS are unchanged. Authors should change the wording regarding this.

·      Authors should look at other adipokines and TGFb in tissues and plasma as well.

·      At the beginning of paragraph 3.6 please highlight better the rationale for the experiments to follow citing literature in support of their importance on the matter.

·      It is not clear how authors explain and speculate about discordant findings of increased respiration and either increased or decreased Δᴪm (line ~495 of manuscript).

·      It would be important that authors clarify which NOX are expressed in which WAT.

·      Authors should briefly define the function of enzymes SOD, CAT, and GPx.

·      Please define what 4- HNE levels are in the discussion.

·      The discussion about NRF2 findings is very poor. Considering that authors don’t see its target genes upregulated it is hard to recollect their findinds and consider their conclusions supported.

Author Response

Responses to reviewer #2

Leonardo Matta et al investigate how chronic exercise acts on adipose tissue, with a specific focus on mitochondrial bioenergetics and antioxidant activities. They conclude that chronic effects of physical exercise ameliorate these aspects in retroperitoneal WAT, together with increasing NRF2 expression and reducing DNA damage. The topic is interesting; however many conclusions are not strongly supported by data. I do not suggest further consideration of the manuscript in this journal.

Answer: We are happy that the reviewer shares our interest in the topic of exercise-induced mechanisms of health. However, we respectfully point out that because we are aware that our study has limitations, we carefully formulated our conclusions based on the data, which are technically clean and meaningful. We respectfully disagree that our conclusions are unfounded.

1 - It is not clear in the abstract why γH2AX and p21 investigation is important.

Answer: Thank you for your comment but we respectfully point out that we highlight the importance in the abstract.  

Line (Abstract - 30-32): “Finally, we found exercise reduces the phosphorylation levels of H2AX histone (γH2AX), a central protein that contributes to genome stability through the signaling of DNA damage.”

2) Please define “hormetic” the first time the term is mentioned.

Answer: Thank you for your comment. In the first paragraph of the introduction, we have rewritten the definition of hormesis to make it clearer to the reader.

Lines (46-50): "Several cellular and systemic effects induced by exercise seem to be related to hormesis [3]. The hormetic response is defined as a beneficial or stimulatory effect caused by exposure to low doses of an agent known to be toxic at higher doses (e.g., chemical or physical), generating adaptive beneficial effects to the physiology of an organism [2,4]."

3)   Provide molecular mechanisms by which improves Δᴪm, -respiration, -fatty acid oxidation, -structural and -functional integrity of the matrix, and -ATP synthesis.

Answer: Thank you for this suggestion. This issue was extremely relevant, and we decided to add these paragraphs below to the discussion:

Lines (584-598): “In our previous study, we demonstrated that one session of aerobic exercise increased respiratory activity, ATP-, and ROS production in mitochondria isolated from retroperitoneal WAT [30]. Although one acute exercise bout acts as a stressor to redox- and bioenergetic homeostasis, the literature is clear in demonstrating the benefit of chronic exposure to physical exercise on mitochondrial function [62–65]. In the present study, chronic aerobic exercise increased mitochondrial respiration, resulting in a higher flux of protons to the intermembrane space, which favors a greater transmembrane potential and a higher driving force for ATP synthase activity. These results demonstrate that exercise provides an adaptive mitochondrial response to produce ATP more efficiently, due to the more negative ∆ᴪm, which was associated with a lower ROS/ATP ratio. Furthermore, among the changes induced by exercise, the increase in intracellular calcium concentration, ROS production, and the AMP/ATP ratio are important mediators for PGC-1α and Transcription factor A mitochondrial (TFAM) activation. These transcription factors play an important role in maintaining mitochondrial integrity, function, and morphology [62,66].”

4) It would be important that authors mentioned or reported any literature about chronic exercise that actually evolves being too much and provokes major oxidative stress.

Answer: Thank you for your comment about the degenerative response to strenuous exercise. However, our study addresses the chronic changes induced by programmed exercise. The deleterious effects induced by exercise are not due to its exposure over the years (chronicity). In fact, the literature is clear in pointing out the role of physical exercise throughout life for the prevention and treatment of various pathologies associated with low-grade inflammation. (1,2)

We believe that the topic you are referring to is strenuous exercise, characterized by high intensity and high quantity (training volume). In fact, there is a limit between healthy and deleterious adaptations. In this case, strenuous exercise is harmful to the body and causes a series of changes in mitochondrial and redox metabolism that can trigger structural damage, inflammation, and, in the worst cases, inflammatory diseases. (3)

While we appreciate your thoughts, we chose not to discuss this topic, as we did not have a group under this strenuous training condition, and, also, as the objective of the study was to evaluate the chronic effect of programmed exercise.

  • McGee, S.L., Hargreaves, M. Exercise adaptations: molecular mechanisms and potential targets for therapeutic benefit. Nat Rev Endocrinol16, 495–505 (2020). https://doi.org/10.1038/s41574-020-0377-1.

  • Febbraio, M. Health benefits of exercise — more than meets the eye!. Nat Rev Endocrinol13, 72–74 (2017). https://doi.org/10.1038/nrendo.2016.218.

  • GABRIEL, Brendan M.; ZIERATH, Juleen R. The limits of exercise physiology: from performance to health. Cell metabolism, v. 25, n. 5, p. 1000-1011, 2017.

5) A major aspect of concerns regards the utilization of only retroperitoneal WAT for their analyses. Although authors briefly explain its importance in the discussion, it will be necessary that authors assess at least some of the parameters studied, in other WATs, especially visceral, since its importance for metabolic health is apparent. Authors could also consider studying some bioenergetics in the liver as well.

Answer: Thank you for your feedback. One of the objectives of studying the retroperitoneal adipose tissue is its composition, function, and its location in the visceral part, and as such an important indicator for metabolic health, which was the main goal of our study. We think it is valid to rely on the retroperitoneal depot for studying the “health” of white adipose tissue. Regarding more analyses in the liver and other adipose tissue depots, within the 10-day time frame provided for revision, we were unable to measure any other tissues. We have pointed this out in the limitations of the study at the end of the discussion.

6) Authors should look at total cellular ROS as well.

Answer: Thank you for your comment. Analysis of ROS production by Nadph Oxidases is performed on protein extract from white adipose tissue and therefore reflects ROS production at total cellular levels. Mitochondria, in turn, were isolated so that we could analyze the effect on their functioning, as well as their specific production of ROS, in order to investigate their efficiency.

7) Figure 2A-B is somewhat unclear. Why this says pyr/mal while in the methods it says glutamate? Moreover, what is the difference between baseline medium between CI and CII? Are the substrates given at baseline? If so, authors should put the nomenclature differently (they should all be shifted) to the previous point of injection on the left. Finally, authors should explain why CI baseline is significantly lower during exercise. What are the major sources for mitochondrial activity in adipose tissue? Does exercise change them? Please cite literature on this regard.

Answer: Thanks for pointing out these issues. It was a typo in the methodology. We corrected it to match the figure. Regarding the difference between the basal mean of complex 1 and complex 2, they were insignificant between the complexes, although we have observed a significant difference between the control and exercised groups in the analysis of complex 1. This can be explained by the greater need for substrates and the increase of mitochondrial respiration in the exercised groups. Regarding substrates, these were given as described in the graph. In the baseline pyruvate/malate was added. 2 minutes later ADP was added. After 3 minutes TMPD and ascorbate were added and 1 minute after FCCP. This way, it was possible to analyze each state of the analyzed complex.

8) The n2 control in the blots of fig3 is confusing. Authors should show more blots. it is really hard to appreciate differences if the second sample is just so low. SDHB this is barely seen in Fig3a. Authors should do native blots of mitochondrial complexes. As the western blots stand so far in the current manuscript, considering the confusing levels in the control, data do not seem very compelling and strong.

Answer: Thanks for the detailed analysis of the blots. We revised the figure and increased the proportion and its exposure for better visualization of the results. The panel formed by n of 2 animals per group aims at the best presentation of the data, but the analysis took place in a total of 6 animals per group.

Reviewer 3 Report

This article evaluates the chronic effects of physical exercise (running exercise sessions on a treadmill for 30 min, 5 days per week for 9 weeks) on WAT redox homeostasis and mitochondrial function. Little is known if this beneficial remodeling is linked to redox homeostasis. Therefore they analyze reactive oxygen species (ROS) generation, antioxidant enzyme activities, mitochondrial function, markers of oxidative stress and inflammation, and proteins related to DNA damage response.

The authors found that WAT from the exercise group presents higher mitochondrial respiration and ATP production, higher ROS/ATP ratio but lower NADPH oxidase activity, protein carbonylation, and lipid peroxidation levels. These results correspond to increased antioxidant enzymatic activity, reduced glutathione/oxidized glutathione ratio, and higher total Nuclear factor erythroid-2, like-2 (NFE2L2/NRF2) protein levels but lower phosphorylation levels of histone H2AX (γH2AX) and p21 proteins, indicating the hormetic remodeling of adipocyte redox balance.

Methods

The chronic aerobic exercise is based on a previous tested protocol?

Why the choice of retroperitoneal WAT?

NADPH Oxidase Activity was done on frozen samples or on fresh samples?

Line 161, please add “fresh” rWAT. It is not completely clear which tests are done on fresh tissue and which ones from frozen samples

Mitochondrial ATP production, described in 2.7.3 was done on mitochondria previous subjected to oxygen consumption measurement protocol, included FCCP treatment?

Lines 286-87 Protein expression quantification was performed by densitometric analysis and analyzed in the ImageJ program.... there is a reason you haven't used Chemidoc and Image Lab? Is it because wb experiments were done by a different lab?

Line 300: what p/mg of protein represent?

Line 328 (Supplementary Figure 1d, e).

Lines 330-332 this sentence is not clear, is referred to Supplementary Figure 1e or f? When you write in the second bout.....to compared before -what are you referring to?

Line 335 this sentence is referred to (Supplementary figure 1f)? Or Supplementary figure 1 g,h?

Figure 1 # is described in the legend Fig as # Difference intra-group, what exactly means? it is not clear if is the statistical difference between 4th and 8th week, or intra animal group

In Fig 1 it should be interest to put beside the QA (Fig 1f), the is increased in exercise group, the total adipocytes numbers obtained from QA and total WAT to exclude adipocyte hypertrophy.

In the description of Figure 2b it should be noted that only one state is significantly increased in exercised group

Lines 368-369 Δᴪm was lower and the ATP production higher in the exercise group compared to the control (Figure 2c,d), do these two results make sense? Also the discussion does not explain the contrast with respect to the increase of the Δᴪm correlated with an increased respiration (line 493)

Lines 370-3: Interestingly, no differences between groups were found for mitochondrial ROS production (Figure 1e). Consequently, the ROS/ATP ratio was lower in mitochondria from the exercise compared to the sedentary group (Figure 2df), and no differences in electron leakage were found between groups (Figure 2eg).

Lines 373-4 Taken together, these results demonstrate an increase in mitochondrial efficiency in mitochondria isolated from WAT of chronically exercised rats, also result of Figure 1c?

Line 378 d e Extramitochondrial ROS production, is this description correct? The letters are not correct (d,d,e,f)

Line 383 This effect was seen in CII (although not significant), CIII, and CV

Lines 393-394 This sentence “The observation of lower lipid accumulation and improved mitochondrial function in WAT suggested to us that these changes could be related to altered redox homeostasis” in this context this statement is somewhat forced and should be supported or formulated hypothetically

Figure 5d, the 3rd line of the blot correspond to 1 exercise animal?, the normalized level should be very high! Data reported in histogram of Figure 5d and others derived from n=6, were n is the number of animals? In the following blot Exercise in indicated by E and not EX, abbreviations must be uniform throughout the text.

Does the absence of change in the expression of antioxidant systems (Figure 6c)contrast with the data reported in fig 4?

Lines 423-4 Sod3 which was lower in the exercise group compared to controls (Figure 6c). This is not correct , according to the symbols shown in the figure, conversely, the expression of gpx3 is reduced in exercise group

Line 440: “not significant” differences were found in IL-6 levels compared to the control group

Lines 456-7 Following the results of Cdkn2a and Cdkn2d mRNA levels with were indifferent between groups (Figure 8g). The sentence is unclear.

The scope of the paper is clear and methods and experimental plan appropriated and well described but the sequence of experiments and results is sometimes not sufficiently motivated and should be made more logical and consequential.

The training rat model should be documented, the impact of oxidative stress and inflammation in WAT metabolism and the link with aerobic chronic exercise should be well documented. In particular the hormetic effect on WAT of exercise-induced ROS production it is not inferred from the experiments and results shown, except for the activation of Nrf2. This point should be better analyzed otherwise the scope of the paper should be referred only to mitochondrial function and exercise induced adipose tissue phenotype protective against oxidative stress in WAT; by analyzing WAT after 9 weeks of exercise it is not possible to analyze the factor that triggers the hormetic response.

Author Response

Responses to reviewer #3

We thank the reviewer for her/his critical feedback.

  • The chronic aerobic exercise is based on a previous tested protocol?

Answer: Thank you for your question. Yes, our protocol of aerobic exercise was based on previous studies about acute response to aerobic exercise, by our group, as well as other evidence in the literature. Here are the references below:

  • Matta, T.S. Fonseca, C.C. Faria, N.C. Lima-Junior, D.F. de Oliveira, L. Maciel, L.F. Boa, A.P.T.R. Pierucci, A.C.F. Ferreira, J.H.M. Nascimento, D.P. Carvalho, R.S. Fortunato, The Effect of Acute Aerobic Exercise on Redox Homeostasis and Mitochondrial Function of Rat White Adipose Tissue., Oxid Med Cell Longev. 2021 (2021) 4593496. https://doi.org/10.1155/2021/4593496.

  • S. Fortunato, D.L. Ignácio, Á.S. Padron, R. Peçanha, M.P. Marassi, D. Rosenthal, J.P.S. Werneck-de-Castro, D.P. Carvalho, The effect of acute exercise session on thyroid hormone economy in rats, Journal of Endocrinology. 198 (2008) 347–353. https://doi.org/10.1677/JOE-08-0174.

  • G. de Araujo, M. Papoti, I.G.M. dos Reis, M.A.R. de Mello, C.A. Gobatto, W.J. Kraemer, Physiological responses during linear periodized training in rats, European Journal of Applied Physiology. 112 (2012) 839–852. https://doi.org/10.1007/S00421-011-2020-2.

  • Why the choice of retroperitoneal WAT?

Answer: Thank you for this question, without a doubt it is important to emphasize the importance of choosing this depot. In the manuscript, we emphasize the following specific condition of this depot, as below:

Lines (565-567): “Compared to other depots, retroperitoneal WAT shows larger adipocytes, higher amounts of lipogenic transcription factors, low expression of fatty acid oxidation genes, which makes it more susceptible to metabolic and energetic challenges [56,57]”.

  • Palou, J. Sánchez, T. Priego, A.M. Rodríguez, C. Picó, A. Palou, Regional differences in the expression of genes involved in lipid metabolism in adipose tissue in response to short- and medium-term fasting and refeeding, J Nutr Biochem. 21 (2010) 23–33. https://doi.org/10.1016/J.JNUTBIO.2008.10.001.
  • Palou, T. Priego, J. Sánchez, A.M. Rodríguez, A. Palou, C. Picó, Gene expression patterns in visceral and subcutaneous adipose depots in rats are linked to their morphologic features, Cell Physiol Biochem. 24 (2009) 547–556. https://doi.org/10.1159/000257511.

  • NADPH Oxidase Activity was done on frozen samples or on fresh samples?

Answer: The samples were immediately frozen in nitrogen and stored at -80°C. The next day the samples were processed, and their activity was evaluated. The extraction, preservation, and analysis protocol is a protocol supported by the literature, routinely performed in different previously published studies.

  • Matta, T.S. Fonseca, C.C. Faria, N.C. Lima-Junior, D.F. de Oliveira, L. Maciel, L.F. Boa, A.P.T.R. Pierucci, A.C.F. Ferreira, J.H.M. Nascimento, D.P. Carvalho, R.S. Fortunato, The Effect of Acute Aerobic Exercise on Redox Homeostasis and Mitochondrial Function of Rat White Adipose Tissue., Oxid Med Cell Longev. 2021 (2021) 4593496. https://doi.org/10.1155/2021/4593496.

  • S. Fortunato, W.M.O. Braga, V.H. Ortenzi, D.C. Rodrigues, B.M. Andrade, L. Miranda-Alves, E. Rondinelli, C. Dupuy, A.C.F. Ferreira, D.P. Carvalho, Sexual dimorphism of thyroid reactive oxygen species production due to higher NADPH oxidase 4 expression in female thyroid glands, Thyroid. 23 (2013) 111–119. https://doi.org/10.1089/thy.2012.0142.

  • CARVALHO, DENISE P. et al. The Ca2+-and reduced nicotinamide adenine dinucleotide phosphate-dependent hydrogen peroxide generating system is induced by thyrotropin in porcine thyroid cells. Endocrinology, v. 137, n. 3, p. 1007-1012, 1996.

  • Osório Alves, J., Matta Pereira, L., Cabral Coutinho do Rêgo Monteiro, I., Pontes dos Santos, L. H., Soares Marreiros Ferraz, A., Carneiro Loureiro, A. C., ... & Marilande Ceccatto, V. (2020). Strenuous acute exercise induces slow and fast twitch-dependent NADPH oxidase expression in rat skeletal muscle. Antioxidants, 9(1), 57.

  • Line 161, please add “fresh” rWAT. It is not completely clear which tests are done on fresh tissue and which ones from frozen samples

Answer: Important consideration, thank you very much. We have added it to the manuscript.

  • Mitochondrial ATP production, described in 2.7.3 was done on mitochondria previously subjected to oxygen consumption measurement protocol, included FCCP treatment?

Answer: Yes, ATP production was analyzed after oxygen consumption. However, ATP production was analyzed only up to 3 minutes after the ADP addition. That is, the ATP produced after this step was not evaluated, not including activation of complex 4 or maximum uncoupled respiration

  • Lines 286-87 Protein expression quantification was performed by densitometric analysis and analyzed in the ImageJ program.... there is a reason you haven't used Chemidoc and Image Lab? Is it because wb experiments were done by a different lab?

Answer: Thank you for your question. We used Chemidoc to develop the membranes, however the Western blotting analysis was made by ImageJ because it is a routine analysis that has already been widely performed by our group.

  • Line 300: what p/mg of protein represent?

Answer: We corrected the writing to make it easier to understand what was verified in the appropriate concentration ranges, as shown below:

Line (348-350): "were performed using an enzyme-linked immunosorbent assay (ELISA) (R&D Systems Inc. kits - Minneapolis, MN, USA) with a detection range of 62.5 - 4,000 pg/mL for TNF-α, and 125.0 - 8,000 pg/mL for IL -6 concentrations".

  • Line 328 (Supplementary Figure 1d, e).

Answer: Thank you, it was fixed.

  • Lines 330-332 this sentence is not clear, is referred to Supplementary Figure 1e or f? When you write in the second bout.....to compare before -what are you referring to?

Answer: Thank you for your detailed observation. It was a mistake, we fixed on the manuscript. It was referred to figure f. About the second bout of the maximal effort test, we are comparing it to the first maximal effort test. It should be presented in figure 1g,h.

  • Line 335 this sentence is referred to (Supplementary figure 1f)? Or Supplementary figure 1 g,h?

Answer: Thank you, fixed it.

  • Figure 1 # is described in the legend Fig as # Difference intra-group, what exactly means? it is not clear if is the statistical difference between 4th and 8th week, or intra animal group

Answer: Very well observed, thank you again for your detailed observations. It was another mistyping, which is now fixed:

# Difference inter-group and * Statistical difference about fourth week.

  • In the description of Figure 2b it should be noted that only one state is significantly increased in the exercised group.

Answer: Thank you very much for pointing this out, we corrected the sentence as per below:

Line (421-431): “Regarding Complex II, mitochondrial O2 consumption in state I was higher, and a tendency of increase in state II was observed in the WAT of the exercise group compared to controls, without significant alterations in mitochondrial complex IV respiration and maximal oxygen uptake of uncoupled mitochondria between groups (Figure 2b). In addition, we observed that ∆ᴪm was more negative and the ATP production higher in the exercise group compared to the control (Figure 2c, d). Interestingly, no differences between groups were found for mitochondrial ROS production (Figure 2e). However, the ROS/ATP ratio was lower in mitochondria from the exercise compared to the sedentary group (Figure 2f), and no differences in electron leakage were found between groups (Figure 2g). Taken together, these results demonstrate an increase in mitochondrial efficiency in mitochondria isolated from WAT of chronically exercised rats.”

  • Lines 368-369 Δᴪm was lower and the ATP production higher in the exercise group compared to the control (Figure 2c,d), do these two results make sense? Also the discussion does not explain the contrast with respect to the increase of the Δᴪm correlated with an increased respiration (line 493).

Answer: We are glad about this question, as it addresses a very important result of our study. In the analysis we performed, we measured how positive or negative the membrane potential is. The membrane potential was more negative, which means the motor drive, as a result of the transmembrane differential is greater. As a result of this increase in the proton motor drive, we observed greater respiration by Complex I and II and an increase in ATP production by ATP synthase, as a result of the generated gradient. We corrected the discussion of the results in the manuscript, thank you very much for this important consideration.

Lines (589-594): “In the present study, chronic aerobic exercise increased mitochondrial respiration, resulting in a higher flux of protons to the intermembrane space, which favors a greater transmembrane potential and a higher driving force for ATP synthase activity. These results demonstrate that exercise provides an adaptive mitochondrial response to produce ATP more efficiently, due to the more negative ∆ᴪm, which was associated with a lower ROS/ATP ratio.”

  • Lines 370-3: Interestingly, no differences between groups were found for mitochondrial ROS production (Figure 1e). Consequently, the ROS/ATP ratio was lower in mitochondria from the exercise compared to the sedentary group (Figure 2df), and no differences in electron leakage were found between groups (Figure 2eg).

Answer: Thank you for your observations. We have already rewritten the mistyping that was done.

  • Lines 373-4 Taken together, these results demonstrate an increase in mitochondrial efficiency in mitochondria isolated from WAT of chronically exercised rats, also result of Figure 1c?

Answer: Excellent question, thank you. Thanks again for pointing this out. As mentioned before, it was a misinterpretation from our side. The membrane potential is more negative, which means that the motor drive as a function of the transmembrane differential is greater. It is related to the increase in respiration capacity in different states of the mitochondrial complexes, resulting in increased production of ATP for the same concentration of ROS, which can be identified as characteristic alterations of increased mitochondrial efficiency (1,2).

1- Bourguignon, A., Rameau, A., Toullec, G., Romestaing, C., & Roussel, D. (2017). Increased mitochondrial energy efficiency in skeletal muscle after long-term fasting: its relevance to animal performance. Journal of Experimental Biology, 220(13), 2445-2451.

2- Mélanie, B., Caroline, R., Yann, V., & Damien, R. (2019). Allometry of mitochondrial efficiency is set by metabolic intensity. Proceedings of the Royal Society B, 286(1911), 20191693.

  • Line 378 d e Extramitochondrial ROS production, is this description correct? The letters are not correct (d,d,e,f)

Answer: Thank you very much. It was rewritten.

  • Line 383 This effect was seen in CII (although not significant), CIII, and CV

Answer: Thank you very much. It was rewritten.

  • Lines 393-394 This sentence “The observation of lower lipid accumulation and improved mitochondrial function in WAT suggested to us that these changes could be related to altered redox homeostasis” in this context this statement is somewhat forced and should be supported or formulated hypothetically

Answer: True, thank you very much for helping with the cohesion of our manuscript. We chose to modify the introductory sentence of this topic, as follows:

Line (462-465): “Since no changes in mitochondrial ROS production were observed, we evaluated other parameters of redox homeostasis. We found lower NOX activity, and higher SOD, GPx, and Catalase activities (Figure 4a-d) in the WAT isolated from the exercise in comparison to the sedentary group.”

  • Figure 5d, the 3rd line of the blot correspond to 1 exercise animal?, the normalized level should be very high! Data reported in histogram of Figure 5d and others derived from n=6, were n is the number of animals? In the following blot Exercise in indicated by E and not EX, abbreviations must be uniform throughout the text.

Answer: Exactly, the third line corresponds to 1 animal from the exercised group. The data expressed as n=6 corresponds to 6 animals per group. We rewrite as (n=6 animals/group), and the abbreviations were and the abbreviations are now correct. Thank you very much for the detailed comments.

  • Does the absence of change in the expression of antioxidant systems (Figure 6c)contrast with the data reported in fig 4?

Answer: Excellent question! Not necessarily, in Figure 4 we evaluated the activity of antioxidant enzyme activity at baseline (72 hours after 9 weeks of exercise intervention). Enzyme activity may be increased and the levels of gene expression related to its expression may not be altered or even decreased due to the increased activity response. It is important to emphasize that alterations in enzymatic dynamics can modulate gene expression, but do not establish a causal relationship. This is an important point raised when it comes to the study of redox homeostasis.

We decided to add to the discussion, as described below:

Lines (646-656): “We did not find any difference in antioxidants-related genes and NRF2 phosphorylation and antioxidants-related gene expression. We attribute these results to the time of tissue collection, as the animals were euthanized 72 hours after the last training session. According to Egan, B., & Zierath, J. R., (2013), an individual exercise bout elicits a rapid, but transient, increase in relative mRNA expression. Alterations in mRNA expression from basal levels are typically observed 3–12 hours after cessation of the exercise and generally return to the basal levels within 24 hours [91]. However, we observed an increase in the total expression of the NRF2 protein. As NRF2 is subject to posttranslational regulation and its protein levels are higher in the exercise group, we attribute that this pathway was previously activated by exercise sessions [92].”

  • Lines 423-4 Sod3 which was lower in the exercise group compared to controls (Figure 6c). This is not correct , according to the symbols shown in the figure, conversely, the expression of gpx3 is reduced in exercise group

Answer: Thank you very much for your critical comments. We looked at the data again and realized that there was an error in plotting the statistical difference. We corrected the glitch and observed that there was an increase in the expression of Gpx2. However, we observed that there was a sample that was characterized as an outlier, leaving the central trend measures, so we chose to exclude the outlier from the group and no statistical differences were observed, including the expression of Gpx3. The figure and the result in the text have been corrected. Your input was important.

  • Line 440: “not significant” differences were found in IL-6 levels compared to the control group

Answer: Thank you, The mistyping was fixed.

  • Lines 456-7 Following the results of Cdkn2a and Cdkn2d mRNA levels with were indifferent between groups (Figure 8g). The sentence is unclear.

Answer: Thank you, it is true. The sentence was rewritten as below:

Line(539-541): “Curiously, these results were not associated with changes in cellular senescence- and cell cycle arrest-related genes. Cdkn2a and Cdkn2d mRNA levels were not different between groups (Figure 8g).

Round 2

Reviewer 2 Report

Manuscript has not been substantially changed. Concerns to reviewer's points after n8 do not seem to have been addressed. My position still stands about lack of novelty and establishment of causal relationships.

Author Response

Manuscript ID: antioxidants-1804375

Matta et al., Exercise Improves Redox Homeostasis and Mitochondrial Function in White Adipose Tissue.

Reviewer #2: The manuscript has not been substantially changed. Concerns about the reviewer's points after n8 do not seem to have been addressed. My position still stands about the lack of novelty and establishment of causal relationships.

Answer: We apologize for any confusion but, in fact, we did reply to all your points as part of our response to the editor, as per the editor’s request (please note that the points are numbered differently in the response to the editors). Thank you for your understanding. Regarding the novelty aspect of our work, there is a lot of evidence about the effects of exercise on redox homeostasis and mitochondrial function of muscle, heart, or liver. However, to the best of our knowledge, we did not find peer-reviewed articles showing the modulation of redox homeostasis, mitochondrial function, and markers of the DNA damage response in response to exercise in rat white adipose tissue. Based on this, we are confident that this work has several novel aspects. Furthermore, we agree with the reviewer that our work is entirely correlative, and any causal relationship within white adipose tissue remains to be proven. As per the editor’s suggestion, in the final part of the discussion of revised version we commented on these limitations. Please find the remaining answers to your previous points for clarification below.

8) The n2 control in the blots of fig3 is confusing. Authors should show more blots. it is really hard to appreciate differences if the second sample is just so low. SDHB is barely seen in Fig3a. Authors should do native blots of mitochondrial complexes. As the western blots stand so far in the current manuscript, considering the confusing levels in the control, the data do not seem very compelling and strong.

Answer: Thanks for the detailed analysis of the blots. We revised the figure and increased the proportion and its exposure for better visualization of the results. The panel formed by n of 2 animals per group aims at the best presentation of the data, but the analysis took place in a total of 6 animals per group. Regarding the expression of complex II, previous articles have shown that in WAT (gonadal and subcutaneous, for example) its expression is weak (Brunetta et al. see below). Regarding the Native blots, it was impossible to address these within the short time frame for revision provided. We appreciate your comments, which have greatly contributed to the improvement of the work, and we hope that the representative figure can be better.

  • Brunetta, H. S., PolitisBarber, V., Petrick, H. L., Dennis, K. M., Kirsh, A. J., Barbeau, P. A., ... & Holloway, G. P. (2020). Nitrate attenuates high fat dietinduced glucose intolerance in association with reduced epididymal adipose tissue inflammation and mitochondrial reactive oxygen species emission. The Journal of Physiology, 598(16), 3357-3371.

9) Fig 5e is confusing. Authors should better specify samples for each lane. Why do they look so different alternatively? Can you expose more?

Answer: Thank you for your question. This explanation was added in the method section (2.9.2):

Lines (257-265): “For an accurate analysis of protein carbonylation, the samples are presented in duplicates. The first line represents the binding of the protein with the DNP antibody, and the second line, which appears unmarked, is the negative control from the same sample that was exposed to the same conditions but in the absence of DNP. The objective of presenting it in this way is to prove through the negative control that the observed signal is specific to the binding of DNP, according to the manufacturer's instruction. We normalized the densitometric values of the dinitrophenylhydrazone (DNP) bands by the endogenous control (β-actin) and then normalized it to the control group [48].”

Below are references to the technique provided by Millipore.

  • Goldbaum, O., Vollmer, G., & RichterLandsberg, C. (2006). Proteasome inhibition by MG132 induces apoptotic cell death and mitochondrial dysfunction in cultured rat brain oligodendrocytes but not in astrocytes. Glia, 53(8), 891-901.
  • Specific inactivation of cysteine protease-type cathepsin by singlet oxygen generated from naphthalene endoperoxides. Nagaoka, Yuki, et al. Biochemical and biophysical research communications, 331, 215-223 (2005).
  • Oxidative stress, induced by 6-hydroxydopamine, reduces proteasome activities in PC12 cells: implications for the pathogenesis of Parkinson's disease. Elkon, H. et al. Journal of Molecular Neuroscience, 24(3), 387-400 (2004).

10) It would appear more appropriate that the authors put the NRF2 part before fig 5.

Answer: Thank you for the consideration, from this perspective, allocating NRF2 data ahead of oxidative damage marker data makes sense. The information becomes more cohesive for the presentation of data related to inflammatory markers. We change the order as requested.

11) Authors should explain why the increase in NRF2 protein does not translate into the high gene expression of antioxidant genes.

Answer: Thank you for this question, which is very important. We attribute gene expression analysis results to the time of tissue collection. The animals were euthanized 72 hours after the last training session. Thus, although we have observed an increase in the total expression of the NRF2 protein and a tendency to increase its phosphorylation 72 hours after the session, it was not possible to observe changes in gene expression because we were observing a chronic effect in a resting state, and not immediately after the session as previously described in our previous work (1). According to Egan, B., & Zierath, J. R., (2013) each bout of exercise is necessary as a stimulus for adaptation, it alone is insufficient to alter the phenotype—training-induced phenotypic adaptation is the consequence of repetition of the stimulus of individual exercise bouts. For a gene upregulated by exercise and training, an individual exercise bout elicits a rapid, but transient, increase in relative mRNA expression of a given gene during recovery. Alterations in mRNA expression several-fold from basal levels are typically greatest at 3–12 hr after cessation of the exercise and generally return to basal levels within 24 hr (2).

  • 1- L. Matta, T.S. Fonseca, C.C. Faria, N.C. Lima-Junior, D.F. de Oliveira, L. Maciel, L.F. Boa, A.P.T.R. Pierucci, A.C.F. Ferreira, J.H.M. Nascimento, D.P. Carvalho, R.S. Fortunato, The Effect of Acute Aerobic Exercise on Redox Homeostasis and Mitochondrial Function of Rat White Adipose Tissue., Oxid Med Cell Longev. 2021 (2021) 4593496. https://doi.org/10.1155/2021/4593496.
  • 2- Egan, B., & Zierath, J. R. (2013). Exercise metabolism and the molecular regulation of skeletal muscle adaptation. Cell metabolism, 17(2), 162-184.

We decided to add to the discussion, as described below:

Lines (551-561): “We did not find any difference in antioxidants-related genes and NRF2 phosphorylation and antioxidants-related gene expression. We attribute these results to the time of tissue collection, as the animals were euthanized 72 hours after the last training session. According to Egan, B., & Zierath, J. R., (2013), an individual exercise bout elicits a rapid, but transient, increase in relative mRNA expression. Alterations in mRNA expression from basal levels are typically observed 3–12 hours after cessation of the exercise and generally return to the basal levels within 24 hours [91]. However, we observed an increase in the total expression of the NRF2 protein. As NRF2 is subject to posttranslational regulation and its protein levels are higher in the exercise group, we attribute that this pathway was previously activated by exercise sessions [92].”

12) Authors should be careful when stating that they observe lower ROS production since they show that mitochondrial ROS are unchanged. The authors should change the wording regarding this.

Answer: We agree that mitochondrial ROS production was not altered. But when we say there was a ROS reduction, we refer to the context of the reduction of NADPH oxidase activity, which was reduced in response to exercise (Figure 4). We revised the text and tried to make this information clearer to the reader.

13) Authors should look at other adipokines and TGFb in tissues and plasma as well.

Answer: Thank you for your suggestion. We appreciate your suggestion and agree that it will be important to understand the mechanisms behind our observations in response to exercise. However, it was impossible to address these within the short time frame for revision provided.

14) At the beginning of paragraph 3.6 please highlight better the rationale for the experiments to follow, citing literature supporting their importance on the matter.

Answer: Thanks for the guidance on improving writing. We chose to better address the idea according to your suggestion. The sentence was rewritten in the text, as below:

Line (454-457): “Improvements on antioxidant defense are generally related to lower DNA damage [52,53]. To determine whether exercise-induced changes in mitochondrial function and redox homeostasis are linked to enhanced DNA protection, we evaluated DNA damage response markers in our experimental groups.”

15) It is not clear how authors explain and speculate about discordant findings of increased respiration and either increased or decreased Δᴪm (line ~495 of the manuscript).

Answer: Thank you for your question. We identified a conceptual mistake in the presentation and discussion of this result. In the analysis we performed, we measured how positive or negative the membrane potential was. The membrane potential was more negative, which means the higher motor drive was favored by the electrochemical gradient. As a result of this increase in the proton motor drive, we observed greater respiration by Complex I and II and an increase in ATP production by ATP synthase, because of the generated gradient.

We corrected the presentation of the results in the manuscript, thank you very much for this important comment.

Lines (506-511): “In the present study, chronic aerobic exercise increased mitochondrial respiration, resulting in a higher flux of protons to the intermembrane space, which favors a greater transmembrane potential and a higher driving force for ATP synthase activity. These results demonstrate that exercise provides an adaptive mitochondrial response to produce ATP more efficiently, due to the more negative ∆ᴪm, which was associated with a lower ROS/ATP ratio.”

16) It would be important that the authors clarify which NOX is expressed in which WAT.

Answer: Thank you for the suggestion. We agree that it is very important to highlight the NOX isoforms of adipose tissue. Therefore, we added this information to the manuscript and detailed that the reduction in ROS production is specific to NADPH and not to mitochondria:

Lines (534-539): “Among the different NOXs isoforms, Nox2 and Nox4 are the most studied in the adipose tissue physiology context. These isoforms are crucial to adipocyte differentiation, and metabolic- and proteostatic homeostatic control of the adipocyte [81–83]. In our study, the exercise did not change Nox 2 and 4 mRNA expression, but decreased NOX activity followed by enhanced activities of the three main antioxidant enzymes SOD, CAT, and GPx in WAT [81–83]”

17)   Authors should briefly define the function of enzymes SOD, CAT, and GPx.

Answer: Thank you for paying attention to this point. We agree that a punctual description of the role of these enzymes is extremely important, we put it in the manuscript as follows:

Line (71-81): “Redox homeostasis improvement induced by exercise is linked to a higher antioxidant defense capacity, which can be generally classified as non-enzymatic and enzymatic. The main enzymes responsible for the enzymatic antioxidant defense are SOD, CAT, and GPx. SOD is a metalloenzyme that catalyzes the dismutation of O2●- into H2O2 and can be found in three isoforms: SOD1 found mainly in the cytosol, SOD2 in mitochondria, and SOD3 in the extracellular space [24]. GPx catalyzes the reduction of H2O2, dependent on the oxidation of GSH to GSSG, releasing H2O [25]. Finally, CAT is a cytoplasmic hemeprotein that catalyzes the reduction of H2O2 to water and O2 [26], taken together these enzymes neutralize pro-oxidative molecules, such as ROS. There are several pieces of evidence supporting the antioxidant effect of exercise in different tissues [27,28]”

18)    Please define what 4- HNE levels are in the discussion.

Answer: We add considerations regarding the importance of measuring 4-HNE in analyzes related to redox homeostasis and oxidative stress, to the discussion as follows below:

Line (568-573): “4-HNE is a product of lipid peroxidation, being one of the most reactive aldehydes [94]. It participates in multiple physiological processes as a nonclassical secondary messenger. Moreover, it forms adducts with cysteine residues of thiol-containing proteins. Its deleterious effects are frequently found to be proportional to the number of adducts formed affecting negatively mitochondrial ATPase activity, oxygen consumption, and even increasing protein damage [95].”

19)  The discussion about NRF2 findings is very poor. Considering that authors don’t see its target genes upregulated it is hard to recollect their findings and consider their conclusions supported.

Answer: Thank you for that question, it is also very important and is related to your points #11.

We attribute gene expression analysis results to the time of tissue collection. It was impossible to observe changes in gene expression because we were observing a chronic effect in a resting state, and not immediately after the session as described in our previous work. Transcriptional activation mechanisms usually occur in acute responses and can last up to 72 hours. As well as the activation of the Nrf2 protein itself. Although we did not observe an expressive increase in its phosphorylation at 72 hours, an increase in total expression in the tissue was observed. Although we observed an increase in protein expression, this does not imply an increase in its activation. We inserted this consideration into the limitation section:

Lines (599-607): “It could contribute to a better understanding of the specific responses of different WAT deposits and tissues to exercise training. Moreover, our measurements were made 72 hours after the last training session to observe the chronic role of exercise without interference from the last session. Analysis in shorter times after the last exercise session (24 and 48 hours) could better clarify the activation of the NRF2 pathway in our model. Finally, although we demonstrated an increase in antioxidant defense and a decrease in oxidative and DNA damage markers after chronic exercise, experiments demonstrating a direct relationship between them were not performed, such as the evaluation of these responses in an exercised group treated with an antioxidant.”

We chose to remove "NRF2" from the title, so the title of the work was: CHRONIC AEROBIC EXERCISE IMPROVES REDOX HOMEOSTASIS AND MITOCHONDRIAL FUNCTION IN WHITE ADIPOSE TISSUE.